# Phosphorylation of TFCP2L1 by CDK1 is required for stem cell pluripotency and bladder carcinogenesis

Jinbeom Heo[1,2,†], Byeong-Joo Noh[3,†], Seungun Lee[1,2,†], Hye-Yeon Lee[1,2], YongHwan Kim[1,2], Jisun Lim[1,2], Hyein Ju[1,2], Hwan Yeul Yu[1,2], Chae-Min Ryu[1,2], Peter CW Lee[1], Hwangkyo Jeong[4], Yumi Oh[4], Kyunggon Kim[4], Sang-Yeob Kim[4], Jaekyoung Son[1], Bumsik Hong[5], Jong Soo Kim[6], Yong Mee Cho[7,*] iD & Dong-Myung Shin[1,2,**] iD

## Abstract

Molecular programs involved in embryogenesis are frequently upregulated in oncogenic dedifferentiation and metastasis. However, their precise roles and regulatory mechanisms remain elusive. Here, we showed that CDK1 phosphorylation of TFCP2L1, a pluripotency-associated transcription factor, orchestrated pluripotency and cell-cycling in embryonic stem cells (ESCs) and was aberrantly activated in aggressive bladder cancers (BCs). In murine ESCs, the protein interactome and transcription targets of Tfcp2l1 indicated its involvement in cell cycle regulation. Tfcp2l1 was phosphorylated at Thr177 by Cdk1, which affected ESC cell cycle progression, pluripotency, and differentiation. The CDK1-TFCP2L1 pathway was activated in human BC cells, stimulating their proliferation, self-renewal, and invasion. Lack of TFCP2L1 phosphorylation impaired the tumorigenic potency of BC cells in a xenograft model. In patients with BC, high co-expression of TFCP2L1 and CDK1 was associated with unfavorable clinical characteristics including tumor grade, lymphovascular and muscularis propria invasion, and distant metastasis and was an independent prognostic factor for cancer-specific survival. These findings demonstrate the molecular and clinical significance of CDK1-mediated TFCP2L1 phosphorylation in stem cell pluripotency and in the tumorigenic stemness features associated with BC progression.

**Keywords** bladder cancer; CDK1; embryonic stem cell; pluripotency; stemness features
**Subject Categories** Cancer; Stem Cells & Regenerative Medicine; Urogenital System

## Introduction

Both embryonic development and homeostasis in adult tissues are regulated by a population of stem cells (SCs) that undergo self-renewal and also give rise to differentiated progenitors to replace lost cells. Transcription factors (TFs) and chromatin regulatory proteins regulate core functions of SCs by maintaining their specific gene expression patterns (Lambert *et al*, 2018). To maintain pluripotency, embryonic SCs (ESCs) express TFs such as octamer-binding protein 4 (Oct-4), homeobox protein NANOG (Nanog), and SRY-box2 (SOX-2) (Kim *et al*, 2008), which are not expressed in differentiated somatic cells. These TFs form the pluripotency "core circuitry" by reinforcing the expression of genes involved in keeping pluripotent SCs (PSCs) in an undifferentiated state and repressing differentiation-inducing transcription. Expression of pluripotency-associated TFs is largely suppressed once organ development is complete. Aberrant activation of genes that contribute to maintenance of the ESC phenotype and rapid proliferation of ESCs in culture, such as signal transducer and activator of transcription-3 (*STAT3*) (Ho *et al*, 2012), embryonic stem cell-expressed RAS (*ERAS*) (Suarez-Cabrera *et al*, 2018), *MYC* (Mahe *et al*, 2018), Krüppel-like factor-4 (*KLF4*) (Hsieh *et al*, 2017), and catenin beta-1 (*CTNNB1*) (Siriboonpiputtana *et al*, 2017), has been frequently observed in tumors. In particular, a recent study using ~12,000 samples of 33 tumor types from The Cancer Genome Atlas (TCGA) resources demonstrated that stemness indices extracted from transcriptomic and epigenetic data from these tumors are associated with oncogenic dedifferentiation and tumor metastasis (Malta *et al*, 2018).

Bladder cancer (BC) is the fourth most common cancer in men in the USA (Siegel *et al*, 2018). Urothelial carcinoma, the most common histological subtype, accounts for > 90% of BC cases, and

1 Department of Biomedical Sciences, Asan Medical Center, University of Ulsan College of Medicine, Seoul, Korea
2 Department of Physiology, University of Ulsan College of Medicine, Seoul, Korea
3 Department of Pathology, Gangneung Asan Hospital, University of Ulsan College of Medicine, Gangneung, Korea
4 Department of Convergence Medicine, Asan Medical Center, University of Ulsan College of Medicine, Seoul, Korea
5 Department of Urology, Asan Medical Center, University of Ulsan College of Medicine, Seoul, Korea
6 Department of Stem Cell Biology, School of Medicine, Konkuk University, Seoul, Korea
7 Department of Pathology, Asan Medical Center, University of Ulsan College of Medicine, Seoul, Korea
*Corresponding author. Tel: +82 2 3010 5965; Fax: +82 2 3010 8493; E-mail: yongcho@amc.seoul.kr
**Corresponding author. Tel: +82 2 3010 2086; Fax: +82 2 3010 8493; E-mail: d0shin03@amc.seoul.kr
†These authors contributed equally to this work

approximately 70–80% of patients with BC present with non-invasive or early invasive (non-muscularis propria-invasive) disease at initial diagnosis (Holger *et al*, 2016). Non-muscle-invasive BC (NMIBC) is not life-threatening, but recurs in 50–70% of cases and progresses to muscle-invasive BC (MIBC) in 15–25% of cases, resulting in a high risk of distant metastasis and death (Holger *et al*, 2016). While clinicopathological features of BC, such as tumor stage and grade, are useful for risk assessment, relatively few tools are available to predict different clinical outcomes in patients with similar clinicopathological features. The high level of clinical and pathological heterogeneity may also explain the high rates of tumor recurrence and poor outcomes of the advanced stages of this disease. Thus, it is crucial to identify novel molecular markers that elucidate the pathogenesis of BC or predict disease prognosis and treatment responses to enable personalized treatments (Seiler *et al*, 2017).

Importantly, molecular programs involved in embryogenesis are frequently upregulated in BC. For example, *STAT3* (Ho *et al*, 2012), *KLF4* (Choi *et al*, 2014), Sal-like protein 4 (*SALL4*) (Kilic *et al*, 2016), *SOX-2* (Zhu *et al*, 2017), and *CTNNB1* (Chan *et al*, 2009) are aberrantly expressed in BCs, and their high expression is associated with tumor progression and poor prognosis. Elucidating the mechanisms that regulate these TFs and signaling pathways in ESCs and BC cells would significantly advance our understanding of SC characteristics as well as the pathogenesis and stemness features of BC.

Transcription factor CP2-like protein 1 (*TFCP2L1*), a member of the CP2 family of TFs, was identified as a Wnt-responsive gene in mouse embryos (Yamaguchi *et al*, 2005). Transient *Tfcp2l1* expression occurs in the inner cell mass of murine blastocysts, with downregulation shortly after implantation (Pelton *et al*, 2002; Guo *et al*, 2010). In this early embryonic developmental stage, *Tfcp2l1* has a central role in maintenance of a naïve state of pluripotency. In human ESCs that have been converted into a naïve-like state by overexpression of *KLF2*, *KLF4*, and *OCT-4*, expression of *TFCP2L1* is upregulated (Hanna *et al*, 2010), and TFCP2L1 has been identified as the missing pluripotency-associated TF in both murine (Martello *et al*, 2013; Ye *et al*, 2013) and human ESCs (Takashima *et al*, 2014).

Unlike typical pluripotency TFs such as Oct-4 and Nanog, TFCP2L1 is expressed in various epithelia of developing and adult organs, especially in the ducts of exocrine glands and the kidney, where it is important for epithelial morphogenesis, functional maturation, and/or homeostasis (Yamaguchi *et al*, 2006; Werth *et al*, 2017). Indeed, 70% of *Tfcp2l1*-deficient mice die < 5 weeks after birth because of hypoplasia of the kidney. However, regulation of expression levels, activity, and post-translational modification (PTM) status of TFCP2L1 during early embryonic development and in adult pathologies such as cancer have not previously been investigated.

In this study, we investigated the expression, regulation, and PTM of TFCP2L1 in relation to protein activity and stemness features in ESCs and BC cells. Our findings from *in vitro* cell culture assays and an *in vivo* xenograft model suggest that phosphorylation of TFCP2L1 by cyclin-dependent kinase 1 (CDK1) represents a novel molecular circuitry for pluripotency in ESCs and also contributes to proliferation, self-renewal, and invasion of BC cells. In BC patients, activation of the CDK1-TFCP2L1 cascade is associated with aggressive high-grade tumors, lymphovascular invasion (LVI), muscularis propria invasion, frequent metastasis to distant organs, and low patient survival rates. Thus, the present study elucidates the role of pluripotency-associated TFCP2L1 in regulating the stemness features of embryonic and BC cells and demonstrates its consequent clinical relevance in bladder carcinogenesis.

# Results

## Tfcp2l1 in murine ESCs binds to proteins related to pluripotency and regulation of the cell cycle

Tfcp2l1 binds to many transcriptional regulators and chromatin-modifying complexes with roles in ESC self-renewal (van den Berg *et al*, 2010). By analyzing the immunoprecipitation (IP) products of FLAG-tagged Tfcp2l1 in murine ESCs (mESCs), we confirmed that Tfcp2l1 interacted with several proteins related to pluripotency (Oct-4, SOX-2, Nanog, and Klf4), the nucleosome remodeling deacetylase complex (including histone deacetylase proteins Hdac1, Hdac2, and Hdac3, and metastasis-associated protein MTA1), and the transformation/transcription domain-associated protein (Trrap)/p400 complex (including RuvB-like 2 (Ruvbl2), histone acetyltransferase KAT5 (Tip60), and DNA methyltransferase 1-associated protein 1 (DNMAP-1); Fig EV1A).

Next, the Tfcp2l1 interactome was further characterized by mass spectrometry analysis of FLAG-tagged Tfcp2l1 IP products (Dataset EV1). Analysis of interactome dataset by MetaCore software indicated that Tfcp2l1-interacting proteins were highly enriched by protein turnover (translation and degradation) and cell cycle (mitosis and cytoskeletal rearrangement) (Fig 1A), corresponding to a gene network containing the *Cdk1* and *Wnt* pathways (Fig 1B). Gene ontology (GO) analysis indicated that proteins related to G2/M phase transition and spindle assembly were highly represented in the Tfcp2l1 interactome (Fig EV1B and C).

The significance of Tfcp2l1 in cell cycle regulation was further highlighted by investigations of the molecular nature of Tfcp2l1 transcriptional targets in mESCs (Chen *et al*, 2008). MetaCore analysis of the Tfcp2l1 chromatin-IP (ChIP)-seq dataset (with a cut-off value of ≥ 0.5 for transcription start site association scores; Dataset EV2) showed that the targets of Tfcp2l1 were characteristically enriched in genes related to a wide range of cell cycle processes (Fig EV1D).

## Tfcp2l1 is phosphorylated at Thr177 by CDK1

PTM fine-tunes the function of TFs, including SOX-2, Klf4, and Oct-4 (Cai *et al*, 2012). IP of either ectopically expressed or endogenous proteins demonstrated that Tfcp2l1 physically interacted with CDK1 (Fig 1C). *In silico* analysis of putative sites of PTM of Tfcp2l1 identified Thr177 as a site of phosphorylation by CDK1 (Appendix Fig S1A). Western blotting of immunoprecipitated Tfcp2l1 identified threonine phosphorylation (Fig 1D), and mass spectrometry identified phosphorylation in the Tfcp2l1 peptide containing Thr177 (Fig 1E). Site-directed mutagenesis of Thr177 (T177A) abolished threonine phosphorylation in Tfcp2l1 (Fig 1F). Inhibition of CDK1 expression with a specific small hairpin (sh) RNA (sh*Cdk1*), or inhibition of CDK1 activity with roscovitine, a pan-specific CDK

inhibitor, significantly reduced the level of threonine phosphorylation in Tfcp2l1 (Fig 1G and H). To detect phosphorylation, we developed a polyclonal antibody specific for Thr177-phosphorylayted Tfcp2l1 (Appendix Fig S2). Western blotting using this in-house antibody revealed a reduction in the level of phosphorylated Tfcp2l1 (p-Tfcp2l1) after treatment with roscovitine or shCdk1 (Appendix Fig S2D). Together, these results show that Thr177 is targeted for phosphorylation by CDK1 in mESCs. Thr177 site is highly conserved in TFCP2L1 proteins from all species examined, suggesting that it is important for TFCP2L1 function (Appendix Fig S1B).

### Tfcp2l1 Thr177 phosphorylation by CDK1 is essential for proliferation and cell cycle progression of ESCs

The biological relevance of Tfcp2l1 Thr177 phosphorylation was examined by measuring the promoter activity of a *Nanog* reporter and a reporter with six tandem repeats of the binding sites for *Oct-4/Sox-2*, which are targets of Tfcp2l1. Exogenous expression of wild-type *Tfcp2l1* resulted in significantly higher promoter activities than endogenous expression of *Tfcp2l1* or exogenous expression of *Tfcp2l1* encoding the T177A substitution (Appendix Fig S1C). In each case, promoter activity was stimulated by *Cdk1* expression.

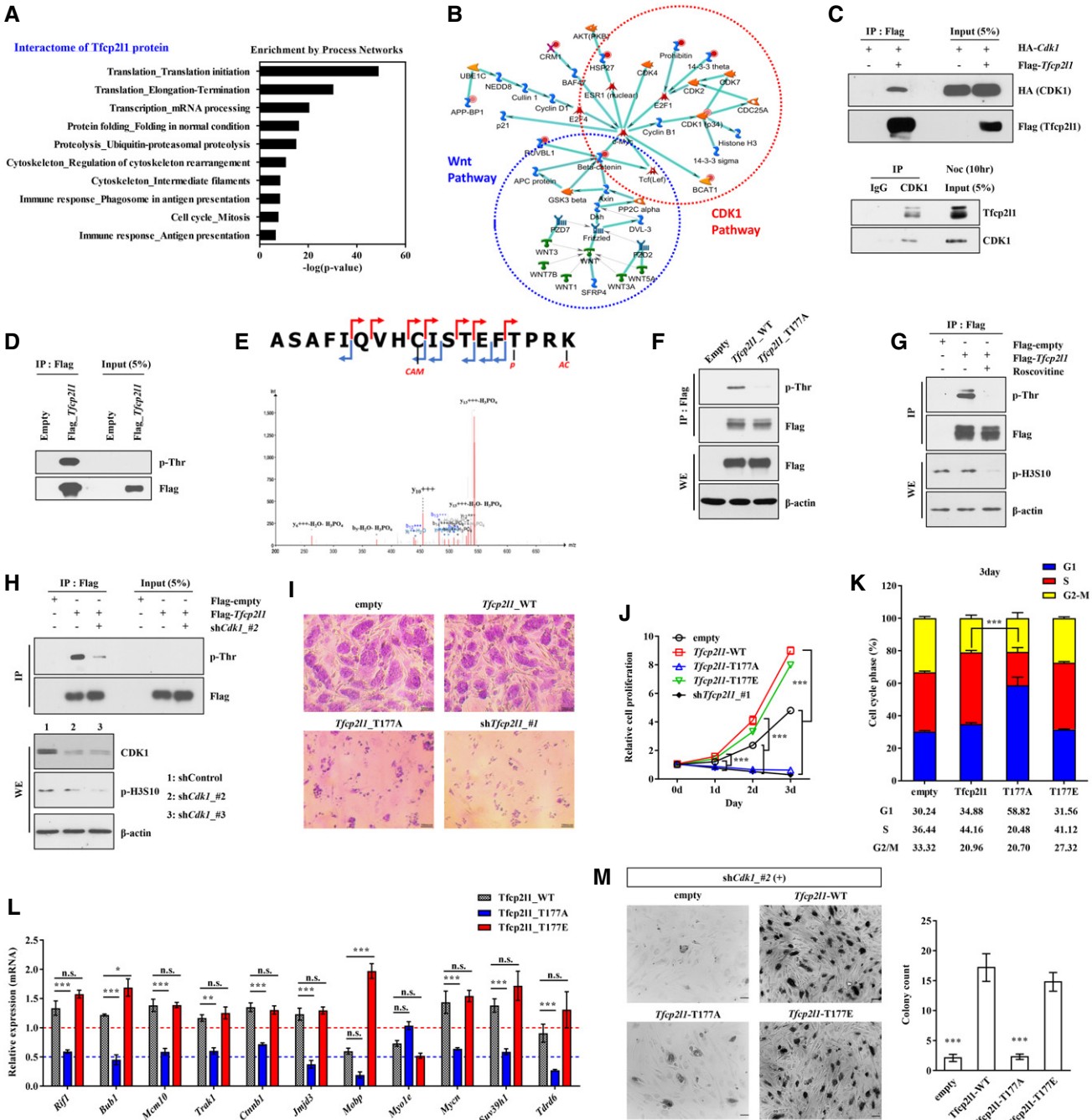

**Figure 1.**

◀

**Figure 1.  Thr177 phosphorylation of Tfcp2l1 by CDK1 is essential for pluripotency and cell cycle progression of mESCs.**

A, B Tfcp2l1 protein interactome, identified by mass spectrometry of IP products in mESCs stably expressing FLAG-tagged Tfcp2l1 (Flag-Tfcp2l1 mESCs). (A) The ten most highly enriched MetaCore Process Networks for the Tfcp2l1 interactome. (B) A representative Gene Network for the Tfcp2l1 interactome associated with the Wnt and CDK1 pathways. The normalized D-score ($D^N$-score) of each interacting protein is indicated by intensity of red coloration.

C IP assay to detect physical interaction between FLAG-tagged (upper panel) or endogenous (lower panel) Tfcp2l1 and CDK1 proteins in mESCs. Protein content of mESCs is shown by lanes containing 5% of the IP input.

D Detection of phosphorylated threonine (p-Thr) in anti-FLAG IP from Flag-Tfcp2l1 mESCs.

E Mass spectrometry of anti-FLAG IP products to detect Thr177-containing peptides. Red and blue lines in the peptide fragmentation map indicate y ions and b ions, respectively. Letter "*p*" and "*AC*" indicate phosphorylation and acetylation, respectively.

F Anti-FLAG IP to detect p-Thr in Flag-Tfcp2l1 mESCs (wild-type or T177A variant).

G Anti-FLAG IP to detect p-Thr in Flag-Tfcp2l1 mESCs in the absence or presence of 25 μM roscovitine for 5 h (to inhibit CDK1).

H Anti-FLAG IP to detect p-Thr in Flag-Tfcp2l1 mESCs with or without transient expression of shRNA for *Cdk1* (sh*Cdk1*). CDK1 activity was assessed by the level of histone H3 phosphorylated at Ser10 (p-H3S10).

I mESCs overexpressing *Tfcp2l1*-WT, *Tfcp2l1*-T177A, or sh*Tfcp2l1* analyzed for alkaline phosphatase (AP) expression (×200 magnification, scale bar = 100 μm).

J–L mESCs overexpressing *Tfcp2l1*-WT, *Tfcp2l1*-T177A, or *Tfcp2l1*-T177E analyzed for cell proliferation (J), cell cycle stages (K), and expression of Tfcp2l1 transcription targets related to the cell cycle by real-time qPCR (L).

M Representative images of AP staining (×200 magnification, scale bar = 200 μm) of sh*Cdk1* mESC colonies rescued by overexpression of *Tfcp2l1*-WT, *Tfcp2l1*-T177A, or *Tfcp2l1*-T177E (left panel) and quantitation (right panel).

Data information: Values are displayed as means ± SEM. *$P < 0.05$, **$P < 0.01$, ***$P < 0.001$ compared with *Tfcp2l1*-WT. Statistical tests used are as follows: one-way (M) and two-way ANOVA (J, K, and L) with Bonferroni *post hoc* tests; n.s. = non-significant. Number of biological replicates is $n \geq 4$. The exact $P$-values and number of replicates can be found in the source data.

Source data are available online for this figure.

Ectopic expression of *Tfcp2l1* enhanced the proliferation capacity of the cells (Appendix Fig S3A) and upregulated expression of G2/mitotic-specific cyclin B and pluripotency-associated TFs (such as Oct-4, Nanog, and SOX-2; Appendix Fig S3B). By contrast, mESCs expressing the *Tfcp2l1*-encoded T177A substitution were severely defective in the establishment of undifferentiated alkaline phosphatase (AP)-positive ESC colonies (Fig 1I). Expression of T177A *Tfcp2l1* in mESCs significantly reduced their proliferation and depleted the number of cells in S phase, starting from day 2 (Appendix Fig S3C) and peaking at day 3 (Fig 1J and K). Accordingly, these mESCs were impaired in the transcription of a subset of Tfcp2l1-targeted genes characterized by the GO term "cell cycle" (Fig 1L). Expression of a Tfcp2l1 phospho-mimic T177E variant increased proliferation and gene expression, as observed with wild-type Tfcp2l1 (Fig 1I–L). mESCs with silenced *Tfcp2l1* expression (sh*Tfcp2l1*-ESCs) exhibited reduced proliferation and expression of pluripotency-associated TFs, cyclin B and cyclin D, and cell cycle-related Tfcp2l1-targeted transcripts (Appendix Fig S3D–G).

Inhibition of *Cdk1* impedes pluripotency and selectively kills ESCs (Huskey *et al*, 2015). Severe knock-down of *Cdk1* (sh*Cdk1*#1) led to rapid cell death (Appendix Fig S3H and I), and mESCs with moderate silencing of *Cdk1* (sh*Cdk1*#2) showed reduced cell proliferation and survival and resembled differentiated cells (Appendix Fig S3I). Ectopic expression of wild-type *Tfcp2l1*, but not the T177A variant, partially prevented cell death and loss of the pluripotency induced by *Cdk1* silencing (Fig 1M and Appendix Fig S3J). A similar rescue effect was observed when T177E Tfcp2l1 variant was expressed. Collectively, these findings demonstrate that phosphorylation by CDK1 is important for Tfcp2l1 function in the maintenance of pluripotency and cell cycle progression of ESCs.

## Inhibition of Thr177 phosphorylation impairs Tfcp2l1 function in mESC differentiation and cell reprogramming

Tfcp2l1 represses the commitment of mouse and human ESCs to multiple lineages (Takashima *et al*, 2014; Liu *et al*, 2017). In an

**Figure 2.  Defective Tfcp2l1 Thr177 phosphorylation impairs the function of Tfcp2l1 in ESC differentiation and cellular reprogramming.**

A Real-time qPCR analysis of pluripotency- and lineage-specific genes in embryoid bodies (EBs). mESCs infected with lentiviruses containing *Tfcp2l1*-WT (wild-type), *Tfcp2l1*-T177A, or *Tfcp2l1*-T177E constructs were used for EB formation. Lentivirus with no inserted coding sequence was used as the control. Expression levels are represented as the ratio to 0-day-old EBs of the control group.

B A schematic overview of the *Oct-4* locus in gc*Oct-4-GFP*, along with the experimental design. DE, distal enhancer; PE, proximal enhancer; PP, proximal promoter.

C Immunostaining of SSEA-1 protein (red) in *Oct-4*[+] (green) germline cells in 7-day-old EBs obtained from gc*Oct-4-GFP* ESCs infected with *Tfcp2l1*-WT, T177A, or T177E lentivirus (×200 magnification, scale bar = 100 μm). A merged image with higher magnification (×400 magnifications, scale bar = 100 μm) is shown in the right panel. Nuclei were counterstained with DAPI (blue).

D Number of GFP[+]/SSEA-1[+] germ cells in the EBs on the indicated days of culture.

E An experimental overview (upper panel) and quantification (lower panel) of primed to naïve reprogramming as determined using epiblast-derived stem cells (EpiSCs) from OG2 mice, which carry the heterozygous Oct-4-GFP (ΔPE) transgene.

F Cell morphology, *Oct-4*-GFP expression, and AP staining (×100 magnification, scale bar = 200 μm) in iPSC colonies generated from *Sox-2*, *Oct-4*, *Klf4*, and c-*Myc* (SOKM)-containing lentivirus-infected OG2-MEFs in the absence or presence of the indicated *Tfcp2l1* ORFs or shRNA lentiviruses.

G, H Quantitation of AP[+] iPSC colonies expressing Tfcp2l1 and its variants (G) or with *Tfcp2l1* silencing (H).

Data information: Values are displayed as means ± SEM. *$P < 0.05$, **$P < 0.01$, ***$P < 0.001$ compared with control group; #$P < 0.005$, ###$P < 0.001$ compared with *Tfcp2l1*-WT. The statistical tests used are as follows: one-way (G, H) and two-way ANOVA (A, D, and E) with Bonferroni *post hoc* tests. Number of biological replicates is $n \geq 3$. The exact $P$-values and number of replicates can be found in the source data.

Source data are available online for this figure.

▶

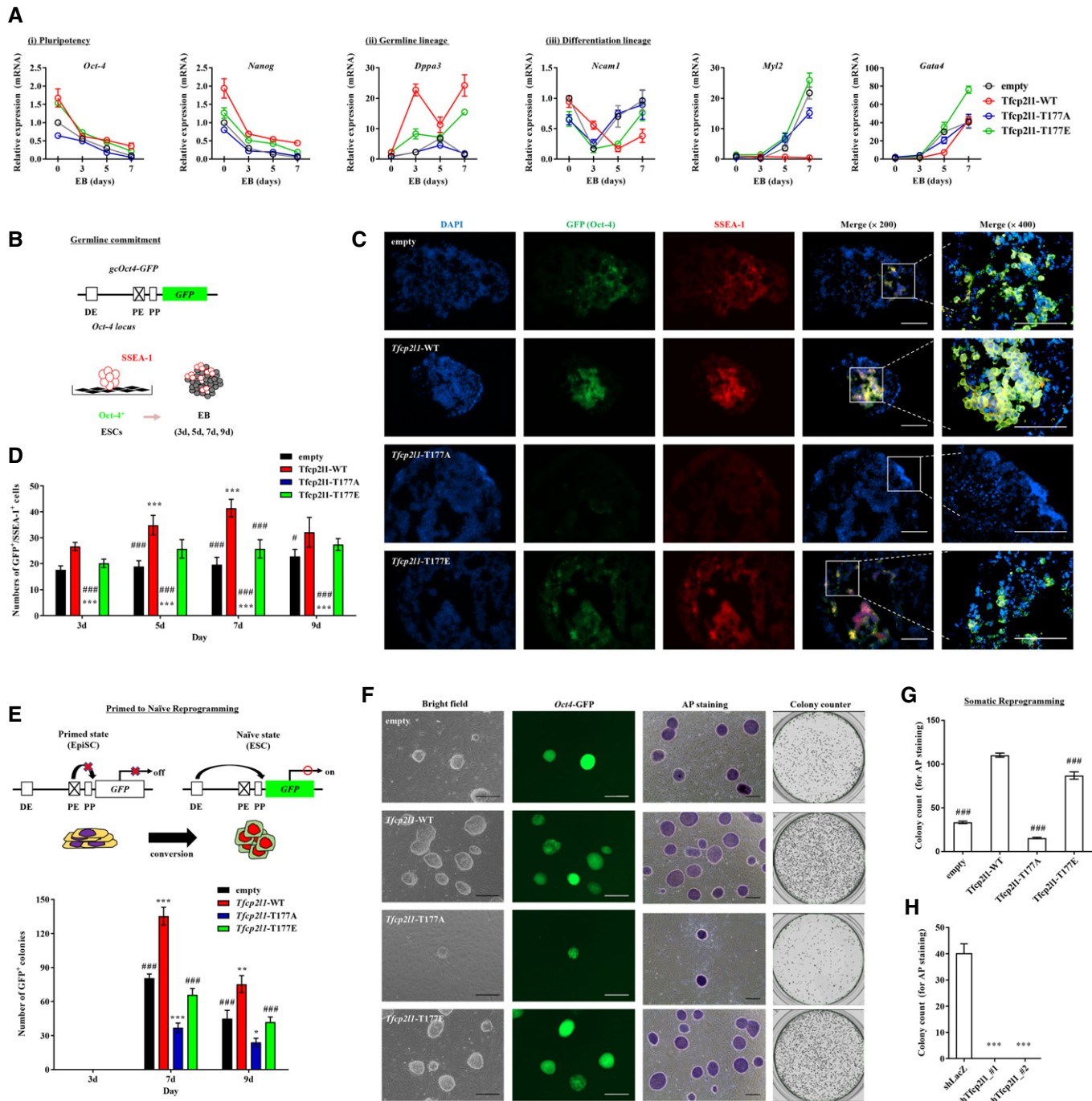

**Figure 2.**

embryonic body (EB)-based differentiation assay with cells stably overexpressing Tfcp2l1 (wild-type or T177A/T177E variants), EB cells derived from ESCs expressing wild-type or T177E Tfcp2l1 showed enhanced expression of genes encoding markers of pluripotency (such as *Oct-4*, *Nanog*, and *Sox-2*) and germ cells (such as *Dppa3*, *Dppa4*, *Ddx4*, and *Sycp3*), and repression of genes associated with differentiation of all three germ layers (such as *Ncam1*, *Myl2*, and *Gata4*; Fig 2A and Appendix Fig S4A) and differentiation mediators including those in the bone morphogenetic protein (BMP), GATA, and inhibitor of DNA binding (ID) families

(Appendix Fig S4B). This expression pattern was not observed in EB cells expressing T177A Tfcp2l1.

The effect of Tfcp2l1 Thr177 phosphorylation on germ cell differentiation was examined by overexpressing *Tfcp2l1* (wild-type, T177A, or T177E variants) in mESCs containing the germ cell-specific marker gcOct-4-green fluorescent protein (GFP; Fig 2B). This marker directs expression of GFP from an *Oct-4* promoter with a deleted proximal enhancer, thereby restricting expression to germ cells during mouse development (Hübner *et al*, 2003). Visualization of primordial germ cells (PGCs) by immunostaining for stage-specific

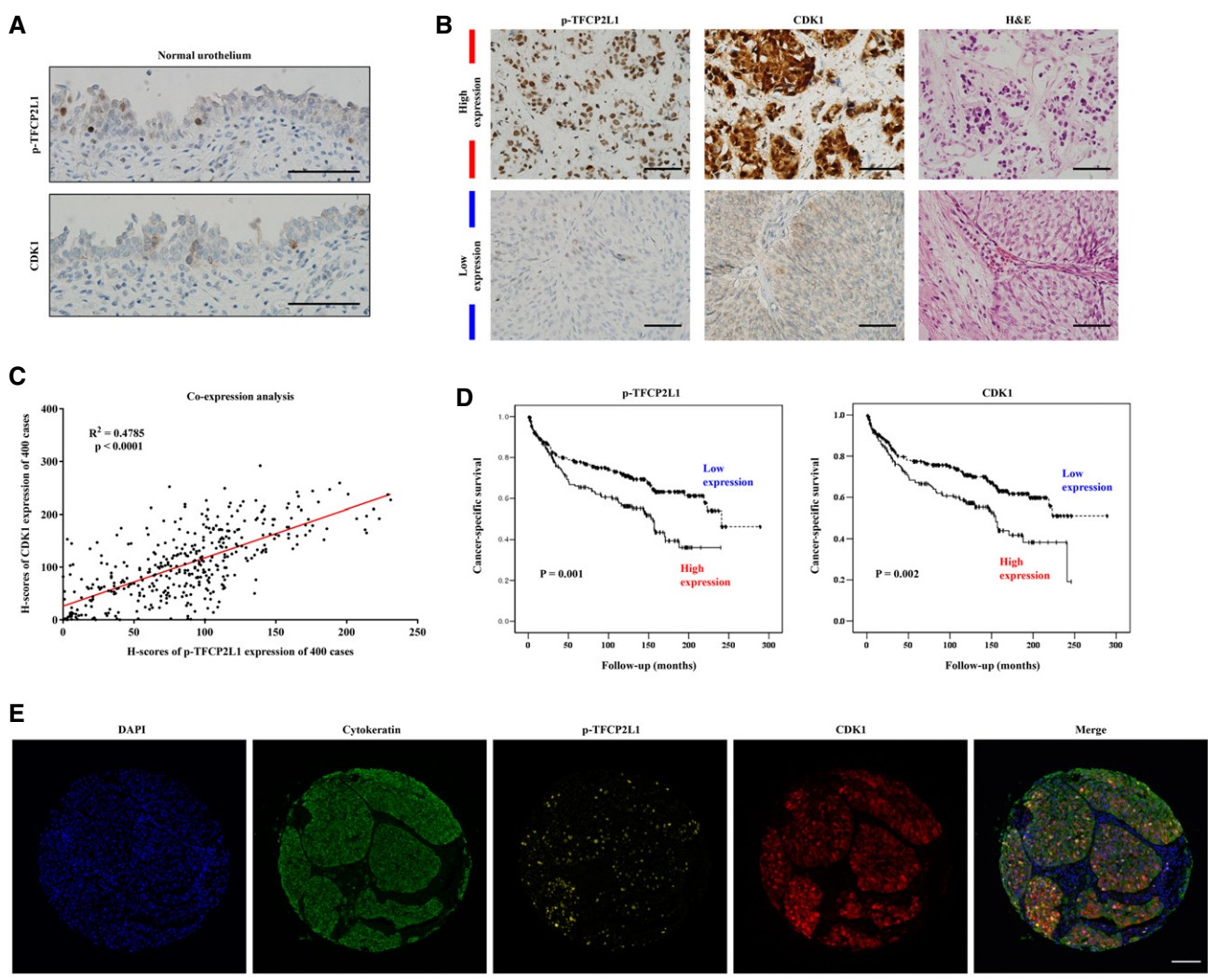

**Figure 3. Upregulation of *TFCP2L1* and *CDK1* in aggressive bladder cancer.**

A    Immunohistochemistry (IHC) staining for p-TFCP2L1 and CDK1 in the urothelium of the normal urinary bladder (×200 magnification, scale bar = 100 μm).

B–E   Expression of p-TFCP2L1 and CDK1 in BC and their correlation with cancer-specific survival. Protein expression was measured by scoring IHC (B, C, D) or Opal multiplex immunofluorescence (IF) staining (E) on a tissue microarray of tumors from 400 BC patients at the Asan Medical Center. (B) Representative images of tumors with high or low expression of p-TFCP2L1 and CDK1, and stained with hematoxylin and eosin (H&E; ×200 magnification, scale bar = 100 μm). (C) Analysis of p-TFCP2L1 and CDK1 expression based on their H-scores. (D) Kaplan–Meier plots of cancer-specific survival according to the expression of p-TFCP2L1 or CDK1 proteins (n = 400). The significance of the differences was assessed by the log-rank test. (E) Representative images of Opal multiplex IF staining of p-TFCP2L1 (yellow), CDK1 (red), and cytokeratin (green) in bladder tumors at ×200 magnification (scale bar = 100 μm). Nuclei were counterstained with DAPI (blue).

Source data are available online for this figure.

embryonic antigen 1 (SSEA-1) after EB formation confirmed the results of previous studies (Heo *et al*, 2017), GFP⁺ SSEA-1⁺ PGCs were identified locally as colonies in EBs derived from ESCs expressing wild-type or T177E Tfcp2l1, or ESCs containing the empty control construct. However, EB cells derived from ESCs expressing T177A Tfcp2l1 did not establish GFP⁺ SSEA-1⁺ PGC colonies (Fig 2C and D; Appendix Fig S4C). These findings demonstrate that Tfcp2l1 Thr177 phosphorylation favors germline lineage differentiation but impedes the multiple somatic lineage commitment of mESCs.

Forced expression of *Tfcp2l1* enhances the reprogramming of naïve pluripotency in primed epiblast SCs (EpiSCs) (Martello *et al*,

2013; Ye *et al*, 2013), and silencing of *Tfcp2l1* inhibits the generation of induced PSCs (iPSCs) (Martello *et al*, 2013). We employed EpiSCs established from OG2 mice (OG2-EpiSCs), which are heterozygous for the *Oct-4-GFP* (ΔPE) transgene (Choi *et al*, 2016; Kim *et al*, 2016), enabling reprogramming to the naïve status to be easily monitored following ectopic expression of *Tfcp2l1* variants (Fig 2E). Expression of T177A *Tfcp2l1* significantly decreased the numbers of reprogrammed (GFP⁺) naïve cells compared with control cells (Fig 2E). The reprogramming of naïve pluripotency was stimulated by wild-type *Tfcp2l1* expression but was little affected by T177E *Tfcp2l1* variant expression. In addition, we infected mouse embryonic fibroblasts (MEFs) established from OG2

Table 1. Expression of p-TFCP2L1 or CDK1 proteins and their co-expression in 400 cases of bladder cancer and their correlation with clinicopathological factors.

| Parameter | p-TFCP2L1 expression | | | CDK1 expression | | | Co-expression proportion | | |
|---|---|---|---|---|---|---|---|---|---|
| | Low (n = 221) | High (n = 179) | P-value | Low (n = 215) | High (n = 185) | P-value | Low (n = 193) | High (n = 207) | P-value |
| Age (years) | | | | | | | | | |
| <70 | 109 (58.6) | 77 (41.4) | 0.209 | 97 (52.2) | 89 (47.8) | 0.55 | 87 (46.8) | 99 (53.2) | 0.582 |
| ≥70 | 112 (52.3) | 102 (47.7) | | 118 (55.1) | 96 (44.9) | | 106 (49.5) | 108 (50.5) | |
| Sex | | | | | | | | | |
| Male | 187 (54.0) | 159 (46.0) | 0.220 | 188 (54.3) | 158 (45.7) | 0.552 | 164 (47.4) | 182 (52.6) | 0.389 |
| Female | 34 (63.0) | 20 (37.0) | | 27 (50.0) | 27 (50.0) | | 29 (53.7) | 25 (46.3) | |
| Size | | | | | | | | | |
| <1 cm | 24 (49.0) | 25 (51.0) | 0.714 | 27 (55.1) | 22 (44.9) | 0.824 | 23 (46.9) | 26 (53.1) | 0.404 |
| 1–2 cm | 83 (55.3) | 67 (44.7) | | 79 (52.7) | 71 (47.3) | | 75 (50.0) | 75 (50.0) | |
| >2 cm | 80 (52.3) | 73 (47.7) | | 77 (50.3) | 76 (49.7) | | 66 (43.1) | 87 (56.9) | |
| Not assessable[a] | 34 (70.8) | 14 (29.2) | | 32 (66.7) | 16 (33.3) | | 29 (60.4) | 19 (39.6) | |
| Multiplicity | | | | | | | | | |
| Unifocal | 118 (59.3) | 81 (40.7) | 0.105 | 114 (57.3) | 85 (42.7) | 0.158 | 105 (52.8) | 94 (47.2) | 0.072 |
| Multifocal | 103 (51.2) | 98 (48.8) | | 101 (50.2) | 100 (49.8) | | 88 (43.8) | 113 (56.2) | |
| Grade | | | | | | | | | |
| PUNLMP | 16 (76.2) | 5 (33.3) | **0.002** | 15 (71.4) | 6 (28.6) | **<0.001** | 16 (76.2) | 5 (23.8) | **<0.001** |
| Low | 73 (62.9) | 43 (37.1) | | 76 (65.5) | 40 (34.5) | | 68 (58.6) | 48 (41.4) | |
| High | 132 (50.2) | 131 (49.8) | | 124 (47.1) | 139 (52.9) | | 109 (41.4) | 154 (58.6) | |
| Lymphovascular invasion | | | | | | | | | |
| Absent | 189 (57.6) | 139 (42.4) | **0.042** | 185 (56.4) | 143 (43.6) | **0.023** | 165 (50.3) | 163 (49.7) | 0.079 |
| Present | 32 (44.4) | 40 (55.6) | | 30 (41.7) | 42 (58.3) | | 28 (38.9) | 44 (61.1) | |
| Carcinoma in situ | | | | | | | | | |
| Absent | 159 (57.6) | 117 (42.4) | 0.157 | 150 (54.3) | 126 (45.7) | 0.721 | 138 (50.0) | 138 (50.0) | 0.296 |
| Present | 62 (50.0) | 62 (50.0) | | 65 (52.4) | 59 (47.6) | | 55 (44.4) | 69 (55.6) | |
| pT category | | | | | | | | | |
| Ta | 68 (62.4) | 41 (37.6) | **0.022** | 65 (59.6) | 44 (40.4) | **0.018** | 58 (53.2) | 51 (46.8) | 0.079 |
| T1 | 94 (57.7) | 69 (42.3) | | 93 (57.1) | 70 (42.9) | | 83 (50.9) | 80 (49.1) | |
| T2–T4 | 58 (47.5) | 64 (52.5) | | 54 (44.3) | 68 (55.7) | | 51 (41.8) | 71 (58.2) | |
| Not assessable[b] | 1 (16.7) | 5 (83.3) | | 3 (50.0) | 3 (50.0) | | 1 (16.7) | 5 (83.3) | |
| Muscularis propria invasion | | | | | | | | | |
| Absent | 162 (60.0) | 110 (40.0) | **0.026** | 158 (58.0) | 114 (42.0) | **0.011** | 141 (51.8) | 131 (48.2) | **0.041** |
| Present | 58 (48.0) | 64 (52.0) | | 54 (44.0) | 68 (56.0) | | 51 (41.8) | 71 (58.2) | |
| Not assessable[b] | 1 (16.7) | 5 (83.3) | | 3 (50.0) | 3 (50.0) | | 1 (16.7) | 5 (83.3) | |
| Lymph node metastasis | | | | | | | | | |
| Absent | 58 (47.2) | 65 (52.8) | 0.911 | 48 (39.0) | 75 (61.0) | 0.267 | 48 (39.0) | 75 (61.0) | 0.866 |
| Present | 25 (48.1) | 27 (51.9) | | 25 (48.1) | 27 (51.9) | | 21 (40.4) | 31 (59.6) | |
| Not assessable[a] | 138 (61.3) | 87 (38.7) | | 142 (63.1) | 83 (36.9) | | 124 (55.1) | 101 (44.9) | |
| Tumor recurrence | | | | | | | | | |
| Absent | 140 (53.8) | 120 (46.2) | 0.442 | 137 (52.7) | 123 (47.3) | 0.563 | 124 (47.7) | 136 (52.3) | 0.761 |
| Present | 81 (57.9) | 59 (42.1) | | 78 (55.7) | 62 (44.3) | | 69 (49.3) | 71 (50.7) | |

**Table 1.** (continued)

| Parameter | p-TFCP2L1 expression | | | CDK1 expression | | | Co-expression proportion | | |
|---|---|---|---|---|---|---|---|---|---|
| | Low (n = 221) | High (n = 179) | P-value | Low (n = 215) | High (n = 185) | P-value | Low (n = 193) | High (n = 207) | P-value |
| Distant metastasis | | | | | | | | | |
| Absent | 198 (56.3) | 154 (43.8) | 0.257 | 197 (56.0) | 155 (44.0) | **0.016** | 176 (50.0) | 176 (50.0) | **0.040** |
| Present | 23 (47.9) | 25 (52.1) | | 18 (37.5) | 30 (62.5) | | 17 (35.4) | 31 (64.6) | |
| Cancer-specific death | | | | | | | | | |
| Alive | 153 (61.7) | 95 (38.3) | **0.001** | 149 (60.1) | 99 (39.9) | **0.001** | 137 (55.2) | 111 (44.8) | **<0.001** |
| Dead | 68 (44.7) | 84 (55.3) | | 66 (43.4) | 86 (56.6) | | 56 (36.8) | 96 (63.2) | |

Values are presented as n (%). Statistically significant parameters ($P < 0.05$) are marked in bold.
PUNLMP, papillary urothelial neoplasm with low malignant potential.
[a]Not assessable because clinical information not available.
[b]Not assessable because of cautery artifact, fragmentation, or incorrect orientation of tumor tissues.

mice (OG2-MEFs) with lentiviruses carrying the TF genes *Oct-4*, *Sox-2*, and *Klf4*, in combination with *Myc*. In this background, expression of wild-type or T177E (but not T177A) Tfcp2l1 enhanced iPSC formation threefold relative to non-treated controls (Fig 2F and G). Silencing of *Tfcp2l1* severely impaired iPSC generation (Fig 2H). Taken together, these findings demonstrate that Tfcp2l1 Thr177 phosphorylation is important for naïve pluripotency and somatic cell reprogramming.

## Tfcp2l1 Thr177 phosphorylation regulates cell cycle-related genes by direct targeting

Because CDK1 drives G2/M transition via its interaction with cyclin B, we examined whether Tfcp2l1 Thr177 phosphorylation occurs in a cell cycle-dependent manner. R1 mESCs synchronized at G2/M phase with nocodazole and released into normal medium to continue the cell cycle showed that Tfcp2l1 Thr177 phosphorylation occurs in a cell cycle-dependent manner, peaking in G2/M phase (Fig EV2A and B). To determine whether Tfcp2l1 gene target binding might be regulated by Thr177 phosphorylation throughout the G2/M phase, we compared Tfcp2l1 binding to a subset of target genes in G2/M-arrested and asynchronized mESCs. Chromatin immunoprecipitation (ChIP) assay revealed that in 47 previously identified Tfcp2l1 binding regions (Chen *et al*, 2008), the relative binding to target chromatin was region-specific (Fig EV2C–E; Appendix Table S1). Furthermore, the binding of Tfcp2l1 to target genes associated with pluripotency and cell cycle process was regulated by Thr177 phosphorylation (Fig EV2F). Low level of Tfcp2l1-T177A binding in mESCs impaired the transcription of genes involved in the cell cycle (Fig 1L) and pluripotency (Fig EV2G and H), but induced several genes that encode differentiation mediators, including BMP, GATA, and ID families (Fig EV2I). These results show that Tfcp2l1 Thr177 phosphorylation is a novel fine-tuning mechanism for cell cycle regulation and developmental processes in mESCs.

## TFCP2L1 phosphorylation is a marker of unfavorable prognosis in urothelial carcinoma

A recent machine learning algorithm study demonstrated that stemness features in pan-cancer patients are associated with oncogenic dedifferentiation and tumor metastasis (Malta *et al*, 2018).

Importantly, the molecular programs of embryogenesis are frequently upregulated in BC (Chan *et al*, 2009; Ho *et al*, 2012; Choi *et al*, 2014; Kilic *et al*, 2016; Zhu *et al*, 2017). This led us to investigate whether TFCP2L1 phosphorylation plays a role in tumorigenesis in epithelium, in which abnormal expression of TFCP2L1 occurs in adulthood (Otto *et al*, 2013; Zaravinos *et al*, 2014). As previously reported (Werth *et al*, 2017), high expression of *TFCP2L1* was observed in kidney and salivary gland among adult tissues and the urinary bladder showed the considerable *TFCP2L1* expression (Fig EV3A). Analysis of open access databases of the Kaplan–Meier plotter (http://kmplot.com/) and TCGA showed that BC and gastric cancer patients with higher *TFCP2L1* expression had shorter overall survival (Fig EV3B–D). Furthermore, analysis of TCGA dataset of BC patients demonstrated that the *TFCP2L1* expression was better associated with shorter overall survival than the SC markers of BC (Fig EV3D).

To investigate the clinical significance of CDK1-mediated phosphorylation of TFCP2L1 (p-TFCP2L1) in BC progression, we used an in-house polyclonal antibody specific for p-TFCP2L1 (Appendix Fig S2). p-TFCP2L1 was specifically detected in a human teratocarcinoma cell line (NTERA2), but not in IMR90 primary fibroblasts, which served as positive and negative controls, respectively (Appendix Fig S2F).

Next, we examined the expression levels of p-TFCP2L1 and CDK1 and their co-expression in normal urothelium of the urinary bladder and in a tissue microarray (TMA) construct generated from transurethral resections of bladder tumor (TURBT) specimens from 400 patients at our institute (Kim *et al*, 2015b). Clinical and pathological characteristics of the 400 patients are shown in Appendix Tables S2 and S3. p-TFCP2L1 and CDK1 were each expressed in a few cells of the basal/parabasal layer of normal urothelium (Fig 3A), with average H-scores of 10.0. By contrast, p-TFCP2L1 and CDK1 were expressed in 92.8 and 93.0% of the bladder tumor samples, respectively (Fig 3B), with a wide range of H-scores (0.0–231.0 with a median of 87.7 for p-TFCP2L1, and 0.0–291.0 with a median of 100.0 for CDK1; Fig 3C).

Notably, p-TFCP2L1 and CDK1 were highly expressed in tumor tissues of BC patients with aggressive clinicopathological features, and this expression was associated with significantly shorter cancer-specific survival (Fig 3D). High levels of p-TFCP2L1 expression were correlated with high tumor grade ($P = 0.002$) and high tumor

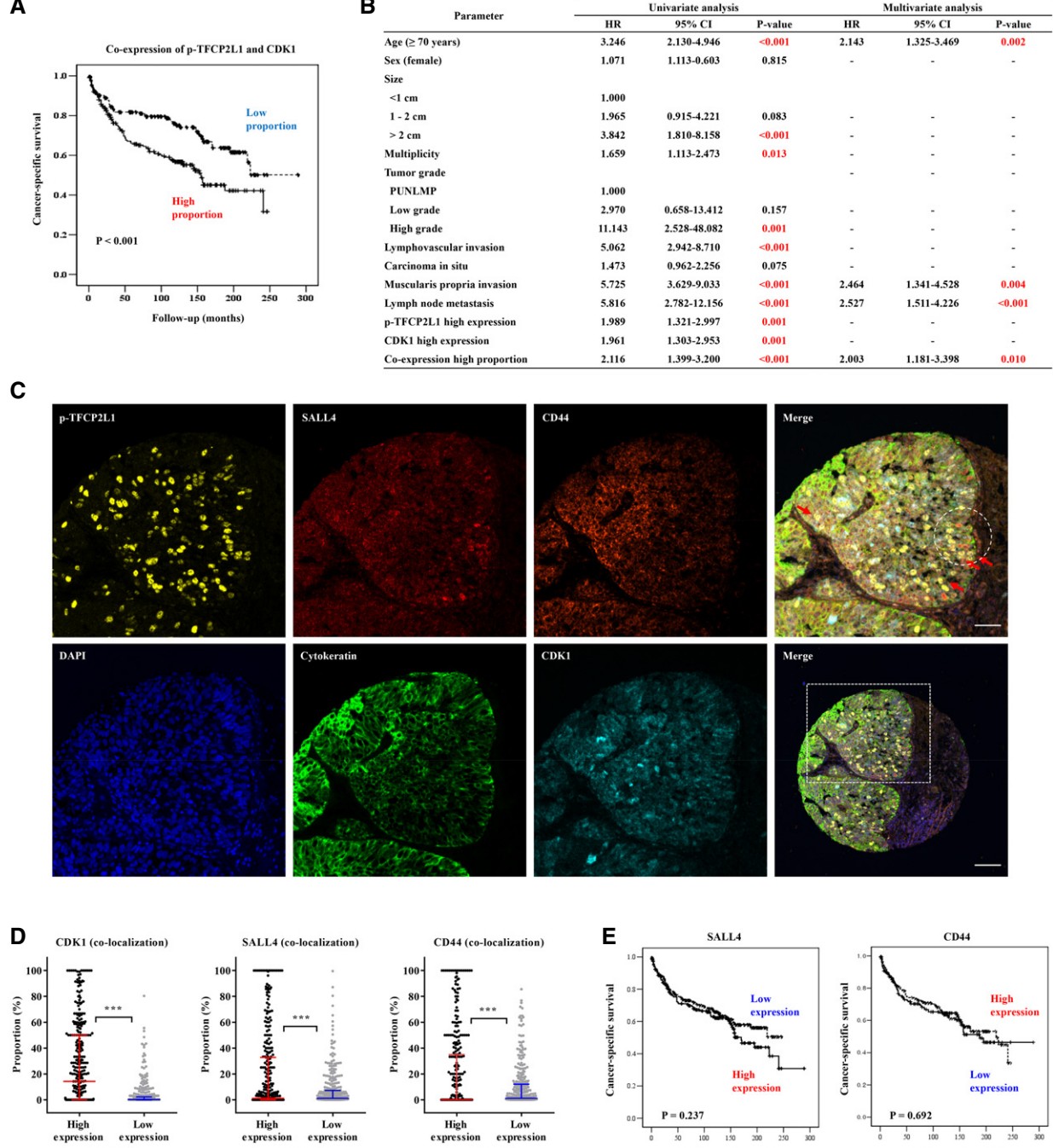

**Figure 4. Co-expression of p-TFCP2L1 and CDK1 is a marker of unfavorable prognosis in urothelial carcinoma.**

A   Kaplan–Meier plots of cancer-specific survival according to the co-expression of p-TFCP2L1 and CDK1 proteins in the BC cohort (*n* = 400). The significance of the differences was assessed by the log-rank test.

B   Univariate and multivariate analysis of clinicopathological factors and expression of p-TFCP2L1 and CDK1 on cancer-specific survival.

C, D   Co-expression of p-TFCP2L1 and BC stem cell markers SALL4 or CD44 by Opal multiplex immunofluorescence (IF) staining in a BC cohort. (C) Representative images of Opal multiplex IF staining for cytokeratin (green), p-TFCP2L1 (yellow), CDK1 (cyan), CD44 (orange), and SALL4 (red) in bladder tumors at ×400 magnification (scale bar = 50 μm). Nuclei were counterstained with DAPI (blue). A merged image with lower magnification (×200) (scale bar = 100 μm) is shown in the right lower panel. (D) Proportion of cells with high or low expression of p-TFCP2L1 that expressed CDK1, SALL4, and CD44, shown as scatter plots with median values and interquartile ranges (*n* = 322, \*\*\**P* < 0.001, unpaired *t*-test).

E   Kaplan–Meier plots and the log-rank tests of cancer-specific survival according to the expression of SALL4 and CD44 proteins on a tissue microarray from 400 BC patients.

Source data are available online for this figure.

stage ($P = 0.022$) as well as with frequent LVI ($P = 0.042$), muscularis propria invasion ($P = 0.026$), and cancer-specific death ($P = 0.001$; Table 1). High levels of expression of CDK1 were correlated with high tumor grade ($P < 0.001$) and high tumor stage ($P = 0.018$), as well as with frequent LVI ($P = 0.023$), muscularis propria invasion ($P = 0.011$), distant metastasis ($P = 0.016$), and cancer-specific death ($P = 0.001$; Table 1).

p-TFCP2L1 expression was positively correlated with that of CDK1 ($P < 0.001$; Fig 3C). We further evaluated the clinical implications of co-expression of these proteins in tumor cells of patients with BC using Opal multiplex immunofluorescence staining, in which tumor cells were identified by cytokeratin staining (Figs 3E and EV4A). High levels of co-expression of p-TFCP2L1 and CDK1 were associated with aggressive clinicopathological features, such as high tumor grade ($P < 0.001$) and frequent muscularis propria invasion ($P = 0.041$), distant metastasis ($P = 0.040$; Table 1), and cancer-specific death ($P < 0.001$; Fig 4A). Co-expression of p-TFCP2L1 and CDK1 was an independent prognostic factor ($P = 0.010$) of cancer-specific survival in multivariate analysis, in addition to age, lymph node metastasis, and muscularis propria invasion (Fig 4B).

Notably, Opal multiplex immunofluorescence staining demonstrated that some p-TFCP2L1-positive cells in bladder tumors highly expressed bladder cancer SC (CSC) markers including SALL4 and CD44 (Figs 4C and EV4B). In cancer cells with high levels of p-TFCP2L1, CDK1, SALL4, and CD44 levels were significantly higher than in cells with low expression of p-TFCP2L1 (Fig 4D), suggesting that a subset of p-TFCP2L1-positive cells could represent a population of bladder CSCs. In line with the results of TCGA dataset analysis (Fig EV3D), p-TFCP2L1 expression (Fig 3D) was more associated with cancer-specific survival than SALL4 or CD44 expression in the BC patient cohort (Fig 4E). Collectively, these results demonstrated that p-TFCP2L1 expression confers higher prognostic significance and clinical utility than that of other CSC markers.

## TFCP2L1 is essential for proliferation and stemness features of human BC cells

To gain mechanistic insights into the clinical relevance of p-TFCP2L1 in bladder carcinogenesis, we first compared the endogenous expression levels of TFCP2L1 and CDK1 between primary human bladder epithelial cells (HBlEpCs) and BC cell lines such as J82 and T24. Compared with HBlEpCs, J82 and T24 cells strongly expressed TFCP2L1 and CDK1 as well as transcription targets such as KLF2 and KLF4 (Fig 5A). HBlEpC proliferation was stimulated by ectopic expression of *TFCP2L1*, enhanced by co-expression of *TFCP2L1* and *CDK1*, and repressed by silencing of these genes (Fig 5B). The positive effect of *TFCP2L1* on cell proliferation was also observed in the T24 BC cell line (Fig 5C and D).

Gene expression profiling analyses of MIBC cell lines have revealed an aggressive basal-like subtype (5637 and HT1197) and a less aggressive luminal-like subtype (HT1376) (Choi *et al*, 2014; Robertson *et al*, 2017). Ectopic expression of *TFCP2L1* stimulated the cell proliferation potency of both subtypes of human MIBC cell lines (Fig 5E), which was significantly inhibited by silencing *TFCP2L1*. Similar results were obtained with the RT4 cell line, a model of NMIBC (Nickerson *et al*, 2017). These results indicate the crucial role of TFCP2L1 in NMIBC and MIBC cells with basal and luminal characteristics.

Next, we investigated whether CDK1 and TFCP2L1 affected stemness features of BC cells by examining tumor sphere-forming and clonogenic capacities. When T24 cells with overexpression or silencing of *TFCP2L1* were seeded on low attachment plates at clonogenic densities, followed by a 1-week culture period, we found that *TFCP2L1*-silenced T24 BC cells hardly produced tumor spheres with sharp edges; however, ectopic expression of *TFCP2L1* or its co-expression with *CDK1* increased tumor sphere formation in comparison with cells transfected with the empty control (Fig 5F). This was also observed in basal-like and luminal-like subtypes of MIBC cell lines, as well as in RT4 NMIBC cells (Fig 5G and EV5A and B). In addition, overexpression of TFCP2L1 and CDK1 in T24 cells increased clonogenic activity in a clonogenic limiting dilution assay (Fig 5H), confirming the importance of CDK1-TFCP2L1 pathways for the stemness features of BC cells.

High levels of co-expression of p-TFCP2L1 and CDK1 were associated with distant metastasis in our cohort of BC patients (Table 1). This prompted us to assess the invasiveness of human BC cells with modified expression of *TFCP2L1* and *CDK1* using a transwell chamber assay. Consistent with clinical results, T24 cells with ectopic expression of TFCP2L1 alone or co-expressed with CDK1 had significantly higher invasion potential than control cells, and silencing of *TFCP2L1* or *CDK1* impaired their invasion ability (Fig 5I).

## TFCP2L1 Thr177 phosphorylation stimulates the stemness features of human BC cells

We next investigated whether functional interplay between CDK1 and TFCP2L1 could be involved in regulating stemness features of human BC cells. Severe knock-down of *CDK1* (sh*CDK1*#1) caused rapid cell death (Fig EV5C and D) and moderate *CDK1* silencing (sh*CDK1*#2) significantly impaired cell proliferation, tumor sphere formation, and clonogenic capacities of T24 BC cells. Importantly, all these defects were significantly rescued by ectopic expression of *TFCP2L1* (Fig 6A–C), which partially restored the level of p-TFCP2L1 (Fig EV5E). These results led us to examine the phosphorylation of TFCP2L1 by CDK1 in BC. Consistent with findings in mESCs (Appendix Fig S2D), *CDK1* knock-down reduced the p-TFCP2L1 level in T24 (Fig 5D) and basal subtype (Fig EV5F) BC cells.

More importantly, IP analysis of FLAG-tagged TFCP2L1 in T24 cells demonstrated that TFCP2L1 was threonine phosphorylated (Fig 6D) and the level of p-TFCP2L1 was dependent on the expression and activity of CDK1 (Fig 6E). Mass spectrometry analysis of the FLAG IPs identified the human TFCP2L1 peptides, which were phosphorylated at Thr177 (Appendix Fig S5). In addition, *in vitro* kinase assay revealed that the Thr177-containing TFCP2L1 peptide, but not the Thr177A-containing peptide was phosphorylated by human CDK1/Cyclin B recombinant proteins (Appendix Fig S6). In line with these findings, TFCP2L1 and CDK1 physically interacted in T24 cells (Fig 6F). Collectively, these findings demonstrate that the CDK1-TFCP2L1 pathway is conserved in human BC cells.

To obtain more direct evidence of the biological relevance of TFCP2L1 Thr177 phosphorylation in BC cells, we constructed Thr177 missense mutations in TFCP2L1 and expressed the mutant proteins in MIBC (T24 and HT1197) and RT4 NMIBC cell lines. Ectopic expression of wild-type TFCP2L1 or the T177E phosphomimic variant stimulated tumor sphere-forming ability (Fig 6G and

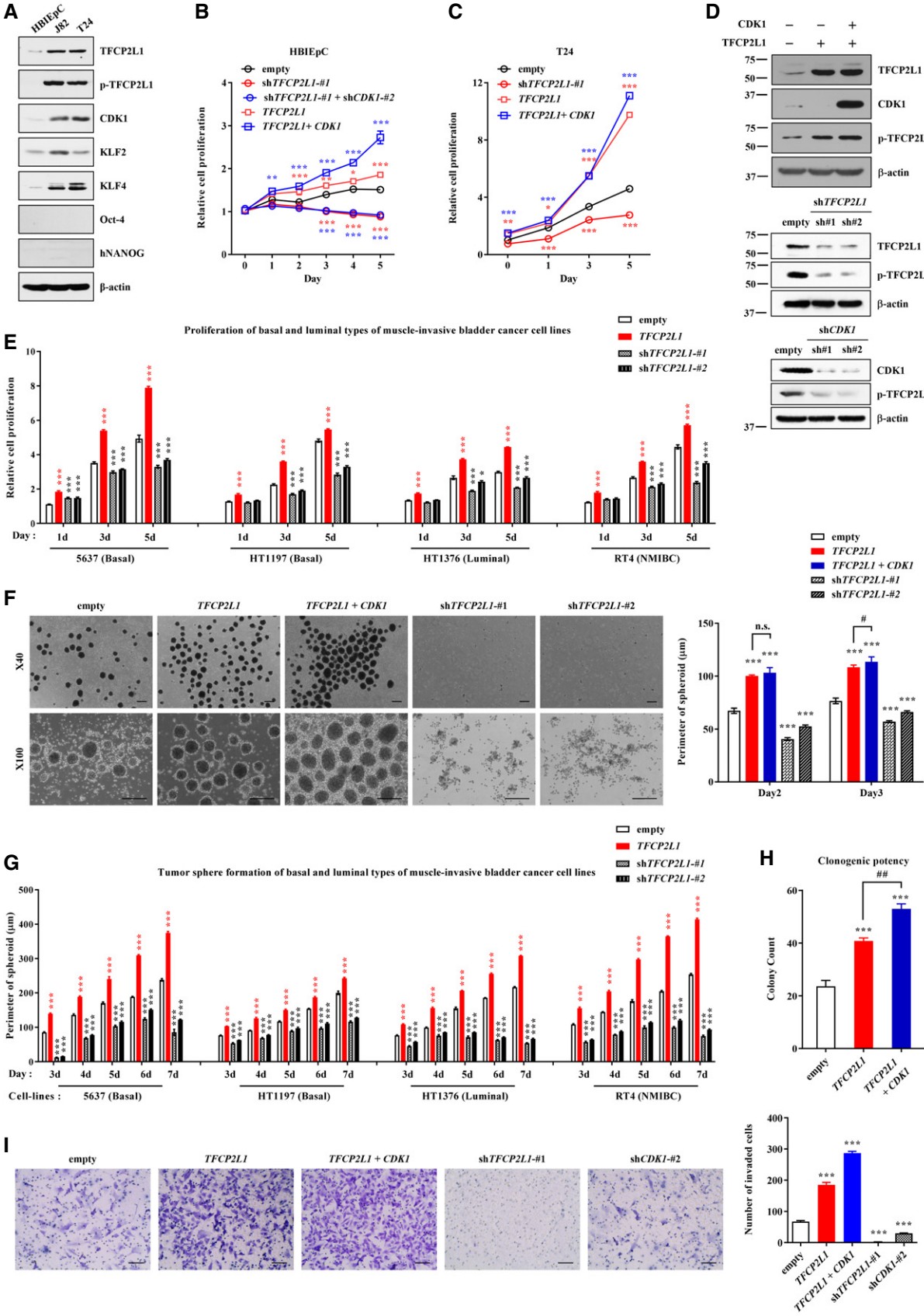

Figure 5.

◀

**Figure 5.  Role of TFCP2L1 and CDK1 on stemness features in human bladder cancer cells.**

A      Western blot analysis of protein expression in primary normal human bladder epithelial cells (HBlEpC) and two human BC cell lines (J82 and T24).

B–E    Cell proliferation of HBlEpC (B), T24 (C), and basal and luminal subtypes of MIBC and NMIBC cell lines (E) after infection with lentiviruses containing human *TFCP2L1* or *CDK1* ORFs or *TFCP2L1* shRNA (two independent shRNAs; #1 and #2). (D) Ectopic expression or silencing of *TFCP2L1* and *CDK1* in T24 cells was validated by Western blot analysis.

F      Tumor sphere formation in T24 cells after *TFCP2L1* silencing, ectopic expression of *TFCP2L1*, or *TFCP2L1* and *CDK1* co-expression. Images are shown at ×40 (upper panel) or ×100 (lower panel) magnification. Scale bars = 200 μm.

G      Tumor sphere formation assay in basal and luminal subtypes of MIBC and NMIBC cell lines with ectopic expression or silencing of *TFCP2L1*. The representative images for each cell line are available as Fig EV5A and B.

H      Clonogenic limiting dilution assay of T24 cells with ectopic expression of *TFCP2L1* or *CDK1* and *TFCP2L1*.

I      Matrigel invasion assays with the indicated T24 cells. Representative images are shown at ×200 magnification. Scale bars = 100 μm.

Data information: All quantitative data are mean ± SEM. *$P < 0.05$, **$P < 0.01$, ***$P < 0.001$ compared with cells transfected with the empty control vector, #$P < 0.05$, ##$P < 0.01$, n.s. = not significant. Statistical tests used are as follows: one-way (H, I) and two-way ANOVA (B, C, E, F, G) with Bonferroni *post hoc* tests. Number of biological replicates is $n \geq 4$. The exact *P*-values and number of replicates can be found in the source data.

Source data are available online for this figure.

H) and clonogenic potential in a limiting dilution assay (Fig 6I) irrespective of the BC cell subtype. Moreover, the transwell chamber assay (Figs 6J and EV5G) revealed that invasion, as well as proliferation and cell cycle progression (Fig EV5H and I), were higher in these cells than in control cells. More importantly, human BC cells expressing the T177A phospho-null variant showed severe defects in stemness features, such as tumor sphere-forming, clonogenic, invasion, and proliferation abilities (Fig 6G–J). Furthermore, forced expression of wild-type or T177E *TFCP2L1* upregulated genes related to cell cycle and stemness; however, T24 cells expressing T177A TFCP2L1 variant or shRNA (sh*TFCP2L1*) showed higher expression of differentiation genes including BMP, ID, and GATA family proteins (Fig EV5J). Taken together, these results illustrate the crucial role of TFCP2L1 Thr177 phosphorylation in generating stemness features of human BC cells.

### TFCP2L1 phosphorylation is essential for tumorigenesis in BCs

To determine the significance of TFCP2L1 Thr177 phosphorylation *in vivo*, we assessed the tumorigenicity of T24 cells overexpressing TFCP2L1 (wild-type or T177A/T177E variants) or shRNA (Fig 7A) by transplanting them orthotopically through the outer layer of the bladder of immunodeficient mice (Kim *et al*, 2017; Ryu *et al*, 2018). When tumor formation was measured 4 weeks after transplantation, T24 cells expressing wild-type or T177E TFCP2L1 exhibited a significantly higher tumor growth than control T24 cells, while cells expressing T177A TFCP2L1 or sh*TFCP2L1* had significantly reduced tumor growth (Fig 7B and C). Histological examination revealed that tumors were well established in lamina propria near the urothelium of bladders in xenografted mice of empty control construct, wild-type, and T177E TFCP2L1 groups; however, they were hardly detected in the T177A TFCP2L1 and sh*TFCP2L1* groups (Fig 7D).

Consistent with the findings of the BC patient cohort (Fig 3B), immunofluorescent staining confirmed that p-TFCP2L1 and CDK1 expression was upregulated in xenograft tumors derived from empty construct-, wild type-, and T177E TFCP2L1-expressing T24 cells (Fig 7E and Appendix Fig S7A), while it was repressed in T177A TFCP2L1 and sh*TFCP2L1* xenografts. In the bladders of sham-operated mice, a few cells of the basal/parabasal layer of urothelium expressed p-TFCP2L1 and CDK1 (Fig 7E). Similar to the immunostaining results of tumor tissues from BC patients (Fig 4C and D), TFCP2L1-expressing cells in xenograft samples showed a high

concurrence with cells stained with the bladder CSC markers including CD44 (Fig 7F), KRT14, and SALL4 (Appendix Fig S8).

Next, we performed an *in situ* proximity ligation assay (PLA) to visualize protein–protein interactions in tissue sections to determine whether p-TFCP2L1 is in physical proximity to CDK1 in tumors generated from xenografts. In control T24-derived tumors, we detected fluorescent signals in the nucleus when p-TFCP2L1 and CDK1 antibodies were used together, indicating an interaction between p-TFCP2L1 and CDK1 (Fig 7G), whereas few signals were detected when only one of the antibodies was used (negative control; Appendix Fig S9A) or in xenograft tumor samples from T177A TFCP2L1 and sh*TFCP2L1* xenografts (Appendix Fig S9B and C), confirming that p-TFCP2L1 interacts with CDK1 in the nuclei of BC cells.

## Discussion

Our results demonstrate that PTM in the CDK1-TFCP2L1 pathway provides a regulatory mechanism that controls the activity of the developmentally crucial TFs, from early embryonic development to tissue homeostasis in adulthood, particularly in bladder epithelium (Fig 7H).

BC is one of the most common urinary malignancies worldwide, with 81,190 new cases and 17,240 deaths in the USA in 2018 (Siegel *et al*, 2018). The high rate of recurrence and distant metastasis of BC, as well as the need for lifetime surveillance and repeat treatments for recurrent disease, has created a huge economic burden (Leal *et al*, 2016). However, molecular targeting therapies are in their infancy, which may be because the molecular mechanisms underlying the clinical and pathological heterogeneity of BCs are unusually complex. Thus, elucidation of the regulatory networks of BC pathogenesis could facilitate the development of novel therapeutics for the treatment of this disease.

Molecular programs of embryogenesis are frequently upregulated during tumorigenesis, including in BC, and upregulation is associated with tumor progression and poor prognosis (Lulla *et al*, 2016). Notably, factors enriched in both ESCs and bladder tumors are well-known members of TFCP2L1 network where they act as upstream regulators, interaction partners, and downstream targets (Dunn *et al*, 2014). Here, we demonstrated upregulation of the CDK1-TFCP2L1 pathway in BCs and showed that it was associated with

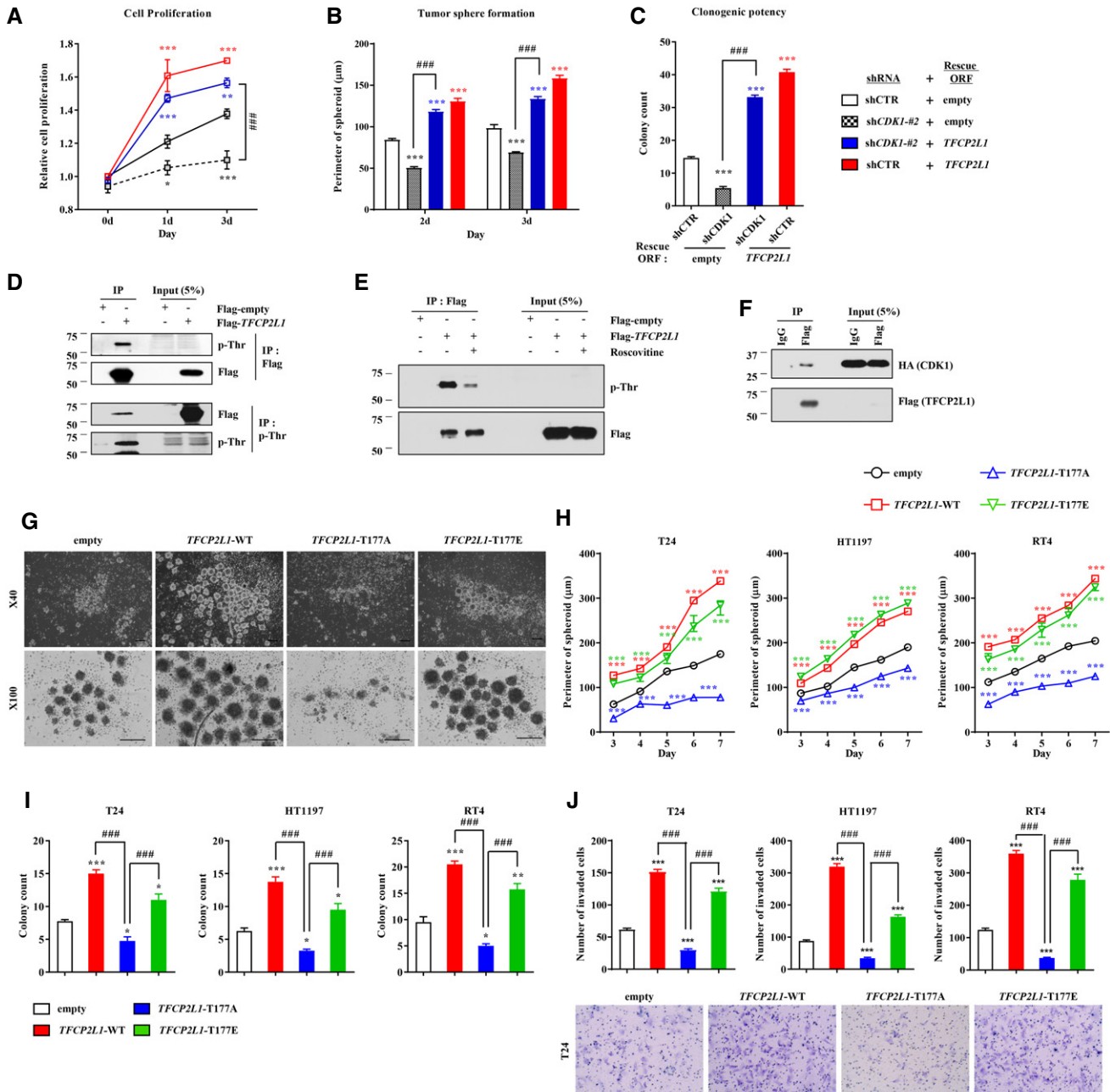

**Figure 6. TFCP2L1 phosphorylation by CDK1 is conserved in human bladder cancer cells.**

A–C Proliferation (A), tumor sphere formation (B), and clonogenic potential (C) assays of *CDK1*-silenced T24 cells with and without expression of *TFCP2L1*.

D–F FLAG IP analysis for the detection of phosphorylated threonine (p-Thr) in Flag-TFCP2L1-expressing cells in the absence (D) or presence of 25 μM roscovitine for 5 h (E), and also for the detection of physical interactions between Flag-TFCP2L1 and CDK1 proteins (F).

G, H Tumor sphere formation (*n* = 6) in T24, HT1197 basal type MIBC, and RT4 NMIBC cell lines infected with lentiviruses containing *TFCP2L1*-WT (wild-type), T177A, or T177E constructs. Lentivirus with no inserted coding sequence (empty) was used as the control. Representative images for the indicated T24 cells are shown at ×40 (upper panel) or ×100 (lower panel) magnification. Scale bars = 200 μm.

I, J Limiting dilution assay for examining clonogenic ability (I) and Matrigel invasion assay (J) of the indicated human BC cell lines. Representative images of T24 cells are shown at ×200 magnification. Scale bars = 100 μm.

Data information: All quantitative data are mean ± SEM. *$P < 0.05$, **$P < 0.01$, ***$P < 0.001$ compared with the control, ###$P < 0.001$, by two-way ANOVA. Number of biological replicates is $n \geq 4$. The exact P-values and number of replicates can be found in the source data.
Source data are available online for this figure.

aggressive clinicopathological features, such as high tumor grade and stage, as well as frequent LVI, muscularis propria invasion, distant metastasis, and cancer-specific death. TFCP2L1-positive cells in bladder tumors displayed high expression levels of SALL4 and CD44 CSC markers. TCGA dataset analysis and our cohort demonstrated that TFCP2L1 status had more prognostic significance than

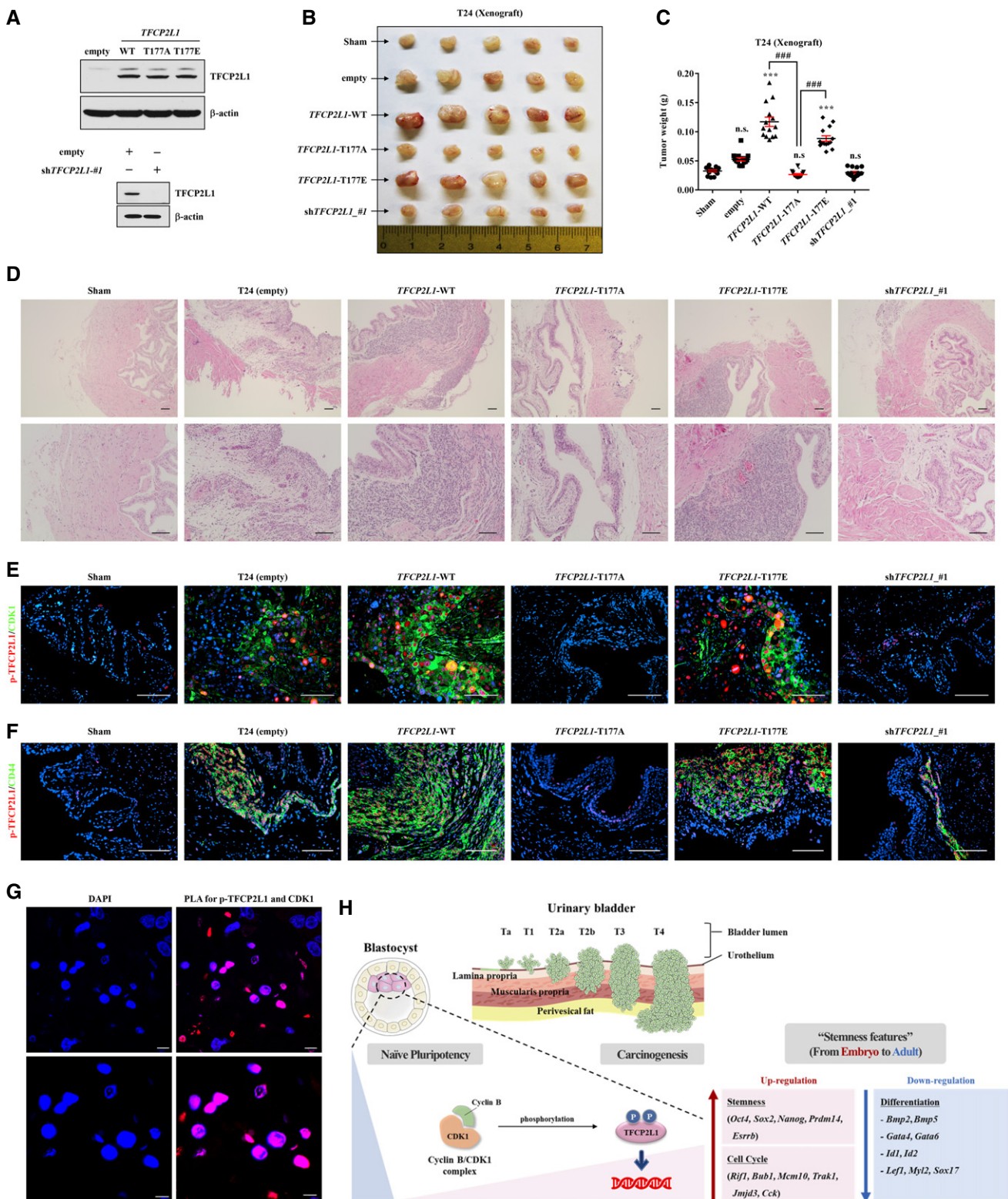

**Figure 7.**

**Figure 7.** *In vivo* significance of TFCP2L1 T177 phosphorylation to the tumorigenesis of human bladder cancer cells.

A Western blot analysis of protein expression in T24 cells infected with lentiviruses expressing *TFCP2L1*-WT (wild-type), T177A, T177E, or shRNA (sh*TFCP2L1*), or infected with the empty construct prior to orthotopic transplantation through the outer layer of the bladder of immunodeficient mice.

B, C Representative images (B) and weight (C) of bladders bearing tumors 4 weeks after transplantation of the indicated T24 cells in triplicate experiments (five mice in each replicate). Data are shown as dot plots of mean ± SEM from fourteen independent animals in each group. \*\*\**P* < 0.001 compared with sham control, ###*P* < 0.001, one-way ANOVA with the Bonferroni post-test.

D Hematoxylin and eosin staining of the bladder tissues of the indicated xenograft groups. Representative images are shown at ×100 (upper panel) or ×200 (lower panel) magnification. Scale bars = 100 μm.

E, F Immunofluorescence assay for detecting p-TFCP2L1 (red) and CDK1 (green; E) or p-TFCP2L1 (red) and CD44 (green; F) in the xenograft tumors. Representative merged images are shown at ×200 magnification. Scale bars = 100 μm. The representative images for the separate fluorescent signals are available as Appendix Fig S7A and B. Nuclei were stained with DAPI (blue).

G Proximity ligation assay (PLA) for detecting colocalization of p-TFCP2L1 and CDK1 in xenograft tumors derived from T24 cells harboring the empty control construct. Little fluorescent signal was observed in the negative control experiments lacking either or both antibodies (see Appendix Fig S9A). Representative confocal microscopic images are shown at ×630 (upper panel) or ×1,000 (lower panel) magnification. Scale bars = 10 μm.

H Schematic overview of our proposed model. Pluripotency-associated TFCP2L1 is regulated by CDK1-mediated PTM; Thr177-phosphorylation of TFCP2L1 by CDK1 regulates its DNA binding to transcription targets related to pluripotency, differentiation, and cell cycle progression. More importantly, the expression and regulatory mechanisms of TFCP2L1 play a critical role in bladder carcinogenesis by regulating the stemness features that are associated with oncogenic dedifferentiation and tumor metastasis in bladder cancers.

Source data are available online for this figure.

that of previously reported CSC markers, suggesting that the CDK1-TFCP2L1 pathway aberrantly triggers the TFCP2L1 network responsible for stemness features in urothelial carcinogenesis. Indeed, the biological significance of TFCP2L1 Thr177 phosphorylation in tumorigenesis of BC was validated in *in vitro* cell cultures and an *in vivo* xenograft model.

In both mESC and BC cell culture models, TFCP2L1 Thr177 phosphorylation by CDK1 promoted cell cycle progression. Aberrations in cell cycle regulation are one of the most extensively studied molecular aspects of BC (Kim *et al*, 2015a; Wang *et al*, 2019). Early cell cycle genes are highly expressed in urothelial carcinomas with a good prognosis. By contrast, urothelial carcinomas subtypes with a poor prognosis display high expression of late cell cycle genes, including CDK1/cyclin B complex and its activators such as CDC25 family genes, and genes related to chromosome segregation and cell division such as *BUB1*, *CDC20*, and *CENP* (Sjodahl *et al*, 2012; Hedegaard *et al*, 2016). In our previous BC cohort study, CDK4 and p27 expression was weakly associated with disease-specific survival (Kim *et al*, 2015b). Accordingly, in the present study, Ki-67, CDK4, and p53 expression, unlike CDK1 expression, only weakly correlated with that of p-TFCP2L1 (Fig EV4E). These results indicate that late cell cycle genes including those of the CDK1-TFCP2L1 pathway have distinct functions in BC pathogenesis. Interestingly, several transcription regulators and enzymes mediating PTM, including acetylation, ubiquitination, and SUMOylation, have been identified as CDK1 substrates (Petrone *et al*, 2016). It is possible that as yet unidentified PTM mechanisms or interacting TF partners may cooperate with p-TFCP2L1, and investigations into their mechanisms could help explain why T177E TFCP2L1 variant has less biological activity than wild-type TFCP2L1. Thus, further research is required not only to explore the mechanism of how the CDK1-TFCP2L1 pathway regulates the stemness features of BC cells outside of its effect on cell cycle progression, but also to identify compounds that target this novel pathway.

In this study, we demonstrated that forced expression of *Tfcp2l1*-T177A in mESCs impaired transcription of susceptible cell cycle genes such as *Rif1* and *Mcm10*, but strongly stimulated expression of embryonic developmental genes involved in the BMP, ID, and GATA pathways. Notably, upregulation of expression of *MCM10* (Li

*et al*, 2016), or repression of urothelial differentiation programs such as the BMP (Shin *et al*, 2014) and GATA (Choi *et al*, 2014) pathways, is associated with tumor progression and poor prognosis of BCs, suggesting that the functional downstream programs of the CDK1-TFCP2L1 pathway might be similar in embryonic and tumor tissues. A recent study also supports this notion. It reported that FOXA1 TF drives enhancer reprogramming during pancreatic ductal adenocarcinoma, thereby triggering an aberrant developmental transition toward embryonic endoderm (Roe *et al*, 2017). However, it is not clear to what extent these mechanisms relate to TFCP2L1, as investigations into the cell type-specific actions of TFCP2L1 have revealed little overlap between the results of Tfcp2l1 ChIP-seq in ESCs (Chen *et al*, 2008) and those in kidney (Werth *et al*, 2017). This lack of overlap might be attributable to the specificity and avidity of antibodies used in the ChIP assays. Therefore, careful and thorough investigation of the functional activities and regulatory circuitry of TFCP2L1 in distinct cell types, developmental stages, and pathological contexts is required.

In BC, treatment selection depends heavily on clinicopathological features, but current clinical staging systems are woefully inaccurate and result in an unacceptably high rate of clinical under-staging and inadequate treatment. The identification of p-TFCP2L1 and CDK1 expression and co-expression as predictive markers of cancer-specific survival could open the prospect of developing p-TFCP2L1/CDK1-based immunohistochemical prognostic markers that could be easily applied in the clinic. In this regard, it will be important to produce several new types of antibodies specific for TFCP2L1. Also, studies are needed to investigate whether a clinical-grade TFCP2L1 antibody would be valuable for the molecular classification of BCs, as well as for the prediction of clinical outcomes from neoadjuvant chemotherapy (Choi *et al*, 2014; Robertson *et al*, 2017).

Carcinoma *in situ* (CIS) represents one of the most important steps toward lethal BC in early-stage disease (Sanli *et al*, 2017). We performed a subgroup analysis on CIS-accompanying cases (Appendix Table S4) and observed a significant association between high co-expression of p-TFCP2L1 and CDK1 in the main tumor and frequent cancer-specific death (*P* = 0.015). Because our cohort did not include pure CIS cases, further studies are needed to examine whether the expression levels of p-TFCP2L1 and CDK1 in pure CIS

cases are also associated with clinical outcomes, such as recurrence-free survival and progression to proper muscle-invasive tumors. In addition, to overcome the limitations of present study regarding its retrospective design and the relatively small number of cases, the significance of CDK1 and TFCP2L1 as prognostic markers needs to be independently validated in separate studies with prospective design and a larger number of BC cases.

In summary, we have shown that phosphorylation of TFCP2L1 at Thr177 by CDK1 is important for ESC pluripotency and cell cycle processes, and that its aberrant activation in adult bladder tissue is associated with tumor progression and poor prognosis of BC. The current findings throw light on how the novel CDK1-TFCP2L1 transcription network involved in normal stemness also acts to modulate stemness features in bladder carcinogenesis in adulthood.

# Materials and Methods

## Study approval

Human BC samples were obtained according to the principles of the Declaration of Helsinki, and the procedures were approved by the Institutional Review Board of the Asan Medical Center (AMC; 2013-107). Written informed consent was received from participants before inclusion in the study. All experiments for preparation of primary mouse cells and xenograft assays were approved by the Institutional Animal Care and Use Committee of the University of Ulsan College of Medicine (IACUC-2016-12-029 and IACUC-2018-12-183).

## Study design

The purpose of this study was to evaluate the molecular, biochemical, and cellular effects of TFCP2L1 Thr177 phosphorylation by CDK1 on ESC pluripotency and cell cycle progression and investigate the clinical relevance of TFCP2L1 phosphorylation for bladder carcinogenesis. *In vitro* cellular studies were carried out using murine R1 and E14TG2a ESCs purchased from ATCC, Manassas, VA. gcOct-4-GFP ESCs were kindly provided by Dr. Hans R. Scholer, Max Planck Institute for Molecular Biomedicine, Münster, Germany, or naïve pluripotency and somatic reprogramming assays. In addition, human primary epithelial cells derived from normal human bladder (HBlEpC; Cell Applications, Inc, San Diego, CA) and human BC cell lines J82, T24, 5637, HT1197, HT1376, and RT4 (purchased from ATCC, Manassas, VA) were employed. *In vivo*, an orthotopic xenograft animal model was employed to determine the effects of TFCP2L1 Thr177 phosphorylation on tumor growth using T24 BC cells harboring missense mutations or shRNA of *TFCP2L1*.

Human specimens in this retrospective study were derived from 400 patients who underwent TURBT between January 1996 and December 2006 at the AMC, and whose tumor tissues were available for TMA construction. Patient clinical information, including tumor recurrence, distant organ metastasis, and survival, was obtained from electronic medical records or hospital charts. All pathological materials, including initial and recurrent tumors, were reviewed for diagnostic reassessment and histological tumor grading according to the 2016 World Health Organization tumor classification (Holger *et al*, 2016). TNM stage was assigned according to the American

Joint Committee on Cancer Staging System, 8th edition (Edge & American Joint Committee on C 2017).

## Orthotopic implantation of BC cells (xenograft)

Male NOD/ShiLtJ-*Prkdc*[em1AMC]*Il2rg*[em1AMC] (NSGA) mice (8 weeks old) were obtained from GEM Biosciences Inc. (Cheongju, Republic of Korea). Mice were housed in the AMC laboratory animal facility in which the temperature was maintained at 21–23°C and 40–60% humidity with a 12-h light–dark cycle (lights on from 8 AM to 8 PM). Mice were maintained group-housed with 2–4 mice per cage and fed with *ad libitum* R/O water and feed (Lab Rodent Chow; Purina, Pembroke, ON, Canada).

Cells were grown in fresh medium 24 h before harvesting with TrypLE (Invitrogen). Harvested cells were washed twice and resuspended in PBS at a concentration of $1 \times 10^7$ cells/ml. Orthotopic implantation of $1 \times 10^6$ cells in a volume of 100 μl was performed by direct injection into the outer layer of the anterior wall and dome of the bladder using a 500 μm syringe and a 26-gauge needle as previously reported (Ryu *et al*, 2018). The mice and site of injection were monitored for 4 weeks. Tumors were recovered by dissection to measure tumor size and perform histological examination or immunostaining with p-TFCP2L1 and CDK1 at day 28 post-injection.

## Cell culture

Murine R1, E14TG2a, and gcOct4-GFP ESCs were grown as previously described (Heo *et al*, 2017). The undifferentiated status of ESCs was assessed using the Alkaline Phosphatase (AP) Detection Kit (Millipore), according to the manufacturer's instructions. The frequency of the AP-stained ESC colonies was analyzed using the GelCount colony counter (Oxford Optronix, Sanborn, NY) at its default settings.

HBlEpC was maintained in Bladder Epithelial Cell Growth Medium (Cell Applications, Inc). Human bladder cancer cell lines J82, T24, 5637, HT1197, HT1376, and RT4 were maintained in Eagle's Minimum Essential (for J82, HT1197, and HT1376), McCoy's 5a Medium Modified (for T24 and RT4), and RPMI-1640 (for 5637) media (ATCC) supplemented with penicillin/streptomycin (Cellgro), and 10% heat-inactivated FBS (Hyclone).

To inhibit activity of CDK1, mESCs or T24 human bladder cancer cell line was treated with 25 μM Roscovitine (Sigma-Aldrich), a pan-specific CDK inhibitor for 5 h before assays.

## TMA construction

TMA blocks with 0.6-mm-diameter cores were constructed from 10% neutrally buffered formalin-fixed, paraffin-embedded urothelial bladder carcinoma tissue from TURBT specimens using a tissue microarrayer (Beecher Instruments, Silver Spring, MD). Three representative cores from different areas of each tumor were included to overcome the issue of tumor heterogeneity. As control cases, normal urothelium was acquired from 14 cases, of which urinary bladder were removed during hysterectomy or colectomy at the AMC for uterine cervical cancer (one case) and colorectal cancer (13 cases), respectively. Clinical and pathological characteristics of the 400 patients are shown in Appendix Tables S2 and S3. Exclusion criteria included unavailable clinical information and cases that were not

assessable because of cautery artifact, fragmentation, or incorrect orientation of tumor tissues.

## Immunohistochemistry (IHC)

IHC staining was performed using an automated staining system (BenchMark XT; Ventana Medical Systems, Tucson, AZ) and an ultraView Universal DAB detection kit (Ventana Medical Systems). The primary antibodies used in this study, their dilutions, and the subcellular location of each antigen are summarized in Appendix Table S5. Nuclei were counterstained with hematoxylin.

## Pathological and IHC assessment

The TMA slides were evaluated by two independent pathologists (B. J. N. and Y. M. C.), both of whom were blind to the associated clinical and pathologic information. IHC and Opal multiplex IHC stains of the TMA slides were analyzed as follows: The percentages of nuclear positivity of p-TFCP2L1 and nuclear and cytoplasmic positivity of CDK1 in cancer cells were recorded, respectively, in each case. The staining intensity was classified according to a four-tiered system: negative (0), weak (1+), moderate (2+), or strong (3+). Then, H-scoring was obtained by multiplying staining intensity of cancer cells (score 0 to 3+) with the percentage of immunoreactive cells, which yielded an H-score ranging from 0 to 300. Additionally, the proportion of cells co-expressing p-TFCP2L1 and CDK1 was recorded, and then, each case was classified as high versus low co-expression based on the cut-off point, which was calculated using receiver operating characteristic (ROC) curve analysis (Fig EV4C and D).

## Opal multiplexed IF staining

Sections of bladder tumor TMA specimens (5 μm) were cut from formalin-fixed paraffin-embedded (FFPE) blocks. Slides were heated for at least 1 h in a dry oven at 60°C and dewaxed using xylene, then dehydrated by sequential incubation in 100, 95 and 70% ethanol, followed by hydrogen peroxide. Antigen was retrieved by microwave treatment (MWT) for 15 min in citrate buffer (pH 6.0). Slides were washed with 1× TBST two times, and blocking was performed with antibody diluent (ARD1001EA, PerkinElmer, Waltham, MA) for 10 min. The first primary antibodies for CDK1 (sc-54; Santa Cruz) were incubated for 1 h in a humidified chamber at room temperature, followed by detection using the Polymer HRP Ms+Rb (ARH1001EA, PerkinElmer) for 10 min. Visualization of CDK1 was accomplished using Opal 690 TSA Plus (dilution 1:50) for 10 min, after which the slide was placed in citrate buffer (pH 6.0) and heated using MWT. In a serial fashion, the slide was incubated with primary antibodies, detected using the Polymer HRP Ms+Rb, and then visualized for CD44 (M7082; Agilent Dako, Santa Clara, CA; Opal 650 TSA Plus), p-TFCP2L1 (in-house; Abfrontier, Seoul, Korea; Opal 620 TSA Plus), SALL4 (CM 384 A,C; Biocare Medical, Pacheco, CA; Opal 570 TSA Plus), SOX-2 (ab92494; Abcam; Opal 540 TSA Plus), and Cytokeratin (AE1/AE3, M3515, Agilent Dako; Opal 520 TSA Plus). Nuclei were subsequently visualized with DAPI (1:2,000), and the section was cover-slipped using HIGHDEF® IHC fluoromount (ADI-950-260-0025, Enzo, USA).

## Quantitative data analysis of Opal staining

Slides were scanned using the PerkinElmer Vectra 3.0 Automated Quantitative Pathology Imaging System (PerkinElmer, Waltham, MA), and images were analyzed using the inForm software and TIBCO Spotfire (PerkinElmer). To acquire reliable unmixed images, representative slides of each emission spectrum and unstained tissue slide were used. Each of the individually stained sections (CDK1-Opal690, CD44-Opal650, p-TFCP2L1-Opal620, SALL4-Opal570, SOX-2-Opal-540, Cytokeratin-Opal520, and DAPI) was used to establish the spectral library of fluorophores required for multispectral analysis. This spectral library formed the reference for target quantitation, as the intensity of each fluorescent target was extracted from the multispectral data using linear un-mixing. Each cell was identified by detecting nuclear spectral elements (DAPI). The total number of CDK1, CD44, p-TFCP2L1, SALL4, SOX-2, and cytokeratin-positive cells was identified and quantified in each tissue.

## Survival analysis using public datasets

For bladder cancer, gene expression dataset of 131 high-grade muscle-invasive urothelial bladder carcinomas in The Cancer Genome Atlas (TCGA; https://cancergenome.nih.gov) was used (The Cancer Genome Atlas Research, 2014). Kaplan–Meier survival analysis was performed using Prism 7.0. Survival analysis of renal cell carcinoma (The Cancer Genome Atlas Research; Creighton et al, 2013; Liu et al, 2018), prostate adenocarcinoma (Taylor et al, 2010; Liu et al, 2018), and esophageal adenocarcinoma (Liu et al, 2018) patients was preformed based on high (red) and low (black) expression levels of TFCP2L1 in two independent cohorts from the TCGA datasets. KM plotter (http://kmplot.com) database were employed with default settings for survival analysis of gastric (Szász et al, 2016), breast (Györffy et al, 2010), ovarian (Győrffy et al, 2012), and lung (Gyorffy et al, 2013) cancer datasets.

## IP pull-down assay

Cell extracts were prepared using IP lysis buffer (50 mM Tris–Cl (pH 7.4), 0.5% NP-40, 150 mM NaCl, 1.5 mM MgCl$_2$, 2 mM DTT, 2 mM EGTA) supplemented with protease/phosphatase inhibitor mixtures (Roche) and then centrifuged (12,000 g for 10 min at 4°C), and IP assay was performed as previously described (Heo et al, 2017).

## Proteomic analysis of the Tfcp2l1 interactome

Tfcp2l1-interacting protein complexes prepared by IP experiments were further reduced with dithiothreitol and alkylated with iodoacetamide prior to digestion with trypsin for 16 h at 37°C. Peptide mixtures were desalted by solid phase extraction using a C18 cartridge and analyzed on an LTQ Orbitrap-XL mass spectrometer (Thermo Scientific) coupled with a nano-LC system (Shimadzu, Kyoto, Japan). Sample processing for mass spectrometry and proteomic data analysis were performed using the CompPASS (Comparative Proteomics Analysis Software Suite) algorithm as previously described (Sowa et al, 2009; Behrends et al, 2010). The

CompPASS scoring metrics are shown in Dataset EV1, and a normalized D-score ($D^N$-score), a representative metric for the identification of candidate interacting proteins, was used for functional analysis of Tfcp2l1 interactome using MetaCore software with the default settings.

### Nano-LC-ESI-MS/MS analysis

To identify the phosphorylated Tfcp2l1 protein, mESCs stably expressing Flag-tagged Tfcp2l1 were used in IP experiments. After SDS–PAGE and Coomassie Brilliant Blue Staining (Bio Rad, Hercules, CA), the phosphorylated band in the IP was analyzed using nano-LC-ESI-MS/MS, which was performed by Diatech Korea (Seoul, Korea), as previously described (Heo *et al*, 2017). Mass spectra were acquired in a data-dependent mode with an automatic switch between a full scan with 5 data-dependent MS/MS scans. The target value for the full scan MS spectra was 30,000 with a maximum injection time of 50 ms. The ion target value for MS/MS was set to 10,000 with a maximum injection time of 100 ms. Dynamic exclusion of repeated peptides was applied for 30 s. For database analysis, raw data from LC-MS analysis were processed using Peptideshaker with the *Mus musculus* sequence database (Uniprot, taxonomy 10090). Digestion enzyme was set to be trypsin with 2 miscleavages option, and precursor ion and fragment ion mass tolerance was set to be 20 ppm and 0.5 Da, respectively. Fixed modifications were set for cysteine (+57.021464 Da: carbamidomethylation). Variable modifications were set for acetylation of K (+42.010565 Da), oxidation of M (+15.994915 Da), and phosphorylation of S, T, and Y (+79.966331 Da), respectively. Fixed modifications during refinement procedure included carbamidomethylation of C (+57.021464 Da), variable modifications during refinement procedure were acetylation of protein N-term (+42.010565 Da), pyrrolidone from E (−18.010565 Da), pyrrolidone from Q (−17.026549 Da), pyrrolidone from carbamidomethylated C (−17.026549 Da). The search engines used X! Tandem, MS-GF+, MS Amanda, MyriMatch, Comet, Tide, Andromeda, OMSSA, Novor, and DirecTag. The parameters of all search engines were set using the default values.

To identify the phosphorylated residue in TFCP2L1 protein, T24 bladder cancer cells stably expressing Flag-tagged TFCP2L1 were used in the IP experiments. The IP sample was digested with chymotrypsin following the filter-aided sample preparation (FASP) method using a Microcon 30k centrifugal Filter unit. The proteins were reduced using 50 mM dithiothreitol in 8 M urea and centrifuged. The eluates were removed, and 200 μl of 8 M urea was pipetted into the filtration unit; the unit was centrifuged again and alkylated using 55 mM iodoacetamide in 8 M urea for 1 h in the dark. The eluates were removed after centrifugation. Additionally, the filter was exchanged three times with 50 mM ammonium bicarbonate buffer. Finally, each sample was treated with 0.1 μg/μl sequencing grade modified chymotrypsin in 50 mM $NH_4HCO_3$ buffer at 37°C overnight. The sample was desalted by solid phase extraction using a C18 cartridge, dried *in vacuo*, and stored at −20°C until further use.

Peptides separation and mass spectrometry analysis were performed as previously described (Kim *et al*, 2018b). Protein identification in human BC cells was conducted against a concatenated target/decoy version of the Homo sapiens complement of the UniProtKB (June 2019). The decoy sequences were created by reversing the target sequences in SearchGUI. The identification settings were as follows: Chymotrypsin, specific, with a maximum of 2 missed cleavages 10.0 ppm as MS1 and 0.5 Da as MS2 tolerances and other sequence database parameters in Peptideshaker were identical as previously described.

### *In vitro* kinase assay

*In vitro* kinase assays were performed by incubating 75 ng CDK1/Cyclin B recombinant human protein (Invitrogen, Waltham, MA) and 100 μg TFCP2L1 wild-type (WT; IQVHCISTEFTPRKHGGEK) or T177A variant (IQVHCISTEFAPRK HGGEK) peptides (Peptron, Daejeon, Korea) in kinase buffer [60 mM HEPES-NaOH pH 7.5, 3 mM $MgCl_2$, 3 mM Na-orthovanadate, 1.2 mM DTT, and 0.5 mM ATP] (Cell Signaling Technology) at 30°C for 15 min. The peptide mixture from the *in vitro* kinase assay was desalted using C18 reverse phase chromatography using ZipTip (Merck, Germany) and dried. The reconstituted peptide mixture containing 0.1% formic acid was introduced into the Ultimate 3000 RSLCnano system (Thermo Fisher Scientific), and the mass to charge ratio of the ionized peptide was measured using Q Exactive Plus Orbitrap mass spectrometry (Thermo Fisher Scientific). LC, mass spectrometry, and database searches were performed as described previously.

### ChIP assay and gene expression analysis

Cross-linked chromatin isolated from cell extracts (from $1 \times 10^7$ cells) was sheared using a Bioruptor Plus sonication device (Diagenode Inc., Denville, NJ) with standard settings (four 20-s pulses with 30-s rest intervals on ice in between), and ChIP analysis was performed using a Magna ChIP G kit (Millipore, Billerica, MA) as previously described (Heo *et al*, 2017).

Quantitative assessment of the mRNA levels of the target genes was performed as described previously (Kim *et al*, 2018a). Total RNA (50 ng) was reverse-transcribed using Taqman Reverse Transcription Reagents (Applied Biosystems, Foster City, CA), and the threshold cycle ($C_t$) was subsequently determined using quantitative PCR (qPCR) as previously described (Jeong *et al*, 2018). The relative expression level of the target genes was determined using the $2^{-\Delta\Delta Ct}$ method, and *Gapdh* was used as the endogenous control gene. Primers used in ChIP and gene expression analysis are listed in Appendix Tables S6 and S7, respectively.

### Western blot analysis

Cell extracts (30 μg) were prepared in RIPA lysis buffer (Santa Cruz Biotechnology, Santa Cruz, CA) supplemented with a protease and phosphatase inhibitor cocktail (Roche, Indianapolis, IN) and separated on 12% SDS–PAGE gels. The expression level of the indicated proteins was assessed by probing with the following antibodies: Tfcp2l1 (OAAB09732; Aviva Systems Biology), phosphorylated Tfcp2l1 (p-Tfcp2l1; home-made; AbFrontier), phosphorylated threonine (p-Thr; #9381; Cell Signaling Technology), Flag-epitope (F3165; Sigma-Aldrich), HA epitope (G046; Abcam), Oct-4 (sc-5279; Santa Cruz), Nanog (ab14959; Abcam), SOX-2 (2683-S; Epitomics), Klf2 (09-820; Millipore), Klf4 (#4038; Cell Signaling Technology),

cyclin A (sc-751; Santa Cruz), cyclin B (sc-752; Santa Cruz), cyclin D (sc-758; Santa Cruz), cyclin E (sc-481; Santa Cruz), CDK1 (sc-54; Santa Cruz), histone H3 phosphorylated at serine-10 (p-H3S10; #9701S; Cell Signaling Technology), Hdac1 (sc-7872; Santa Cruz), Hdac2 (sc-7899; Santa Cruz), Hdac3 (sc-11417; Santa Cruz), Mta1 (#5646S; Cell Signaling Technology), Mbd3 (#3896S; Cell Signaling), Ruvbl2 (#12668S; Cell Signaling Technology), Tip60 (sc-25378; Santa Cruz), DNMAP-1 (10411-1-AP; Proteintech), Trrap (#3966S; Cell Signaling Technology), pCAF (sc-8999; Santa Cruz), and β-actin (A5441; Sigma-Aldrich).

### Embryoid body (EB) or tumor sphere formation

mESCs used in differentiation were maintained with 2i-LIF medium, ESGRO-2i Supplement Kit (Millipore, Billerica, MA) on a 0.1% gelatin coated tissue culture dish. EB formation and *in vitro* germ cell differentiation were performed as previous described (Heo *et al*, 2017). For tumor sphere formation, single cell suspension of tumor cells was resuspended in 1:1 ratio of serum-free Keratinocyte Growth Media (Gibco, Waltham, MA) and Growth Factor Reduced Matrigel (BD Biosciences, Mountain View, CA) and then plated into Ultra Low Attachment plates (Costar, Corning, NY). Tumor sphere formation was assayed 7 days after first plated. Size of EB or tumor spheres was quantified by meaning perimeter from eight randomly chosen representative areas selected from each group using ImageJ software (National Institute of Mental Health, Bethesda, MD).

### *In vitro* cell invasion and limiting dilution assay

Cells were plated at $2 \times 10^4$ cells/well in 100 μl of serum-free DMEM in the upper chamber of transwell permeable supports (Corning Inc, Corning, NY) with 8.0 μm pore polycarbonate membrane filter that was precoated with Matrigel (BD Biosciences) diluted at the ratio of 1:5. The lower chambers were filled with culture DMEM supplemented with 3% FBS. After culturing them at 37°C in a 5% $CO_2$ incubator for 24 h, migrated cells on the lower surface of the membranes were completely removed by using a cotton swab and then were fixed with 4% paraformaldehyde for 10 min and stained with 0.5% crystal violet (Sigma-Aldrich). The cell invasion ability was assessed by counting the number of cells that had migrated to the lower side of the membrane. Quantitative analysis was performed from three randomly chosen visual fields (magnification, ×200) in each transwell chamber.

For limiting dilution assay, BC cells were diluted into a cell density of 1 cell per well and plated into 96-well plate in 50 μl of the culture media. With adding fresh culture media every 2 days, the plated cells were cultivated until 10 days after plating and the number of the colonies was calculated for quantification analysis.

### Immunostaining

For immunocytochemistry, mESCs were fixed with 4% paraformaldehyde (Sigma-Aldrich) for 1 h and co-stained using anti-Oct-4 mouse IgG monoclonal antibody (Millipore) and in-house (Abfrontier, Seoul, Korea) or commercially available (Aviva Systems Biology, clone # OAAB09732) anti-Tfcp2l1 rabbit IgG polyclonal antibodies. IF staining was visualized using Alexa 488- or 564-conjugated anti-mouse or anti-rabbit antibodies (Molecular Probes,

### The paper explained

#### Problem

Aberrant activation of pluripotency-associated genes is frequently observed in tumors, and stemness features are associated with oncogenic dedifferentiation and tumor metastasis, resulting in disease progression, high tumor recurrence, and poor patient survival. Several pluripotency-associated transcription factors such as TFCP2L1, KLF2, and KLF4 are commonly expressed in both early embryonic tissues and a subset of adult tissues. However, the regulatory mechanisms of these pluripotency factors, their signaling pathways in different developmental stages, and their clinical relevance have not been investigated in depth.

#### Results

By analyzing the transcription targets and interacting proteins of Tfcp2l1, a pluripotency-associated transcription factor of murine embryonic stem cells (ESCs), and performing multiplex immunostaining of bladder cancer specimens from 400 patients, we demonstrate that Thr177 phosphorylation of TFCP2L1 by CDK1 is critical for embryonic stem cell (ESC) pluripotency in embryos as well as bladder carcinogenesis in adults. In murine ESCs, this phosphorylation controlled TFCP2L1 binding to targets related to cell cycle and differentiation processes. The physical and functional interaction between TFCP2L1 and CDK1 is conserved in human bladder cancer cells and modulates their proliferation and stemness features. Functional significance of TFCP2L1 Thr177 phosphorylation in bladder carcinogenesis was validated in *in vitro* cell cultures and *in vivo* xenograft models. Moreover, high co-expression of TFCP2L1 and CDK1 in tumor tissues of bladder cancer patients was associated with unfavorable clinical features, including high tumor grade, lymphovascular and muscularis propria invasion, and distant metastasis, and was an independent prognostic factor for cancer-specific survival.

#### Impact

The study provides the first direct experimental evidence that CDK1-mediated phosphorylation of TFCP2L1 affects ESC function and promotes bladder cancer progression. Importantly, the results offer insights into the regulatory mechanisms, expression, and functional profiles of TFCP2L1, demonstrating again the key role of stemness-related transcription factors in the modulation of the stemness features of cancer cells. Bladder cancer, one of the most common urinary malignancies worldwide, is characterized by a high rate of recurrence, and limited targeted therapies are currently available, leading to a huge social and medical burden and an urgent need for novel therapeutic agents. This study identifies TFCP2L1 as a novel molecular marker of bladder cancer that could help (i) elucidate further its pathogenesis, (ii) predict disease prognosis and treatment response to enable personalized medicine, and (iii) facilitate the development of novel therapeutics for the management of this aggressive disease.

Grand Island, NY). To detect the germline committed cells, the EB spheres that formed from gcOct4-ESCs in the indicated days were fixed with 4% paraformaldehyde (Sigma-Aldrich) for 24 h, embedded into paraffin blocks, and cut into 3-μm sections using a microtome. The GFP$^+$ (green) germ cells were further analyzed by immunofluorescence (IF) staining using anti-SSEA-1 mouse IgM monoclonal antibodies (DSHB, Iowa City, IA) and visualized using Alexa 564-conjugated anti-mouse IgM antibodies (Thermo Fisher Scientific, Waltham, MA). Nuclei were counterstained with 4′,6-diamino-2-phenylindole (DAPI, Sigma-Aldrich). The stained samples were photographed using an inverted fluorescence microscope (EVOS® FL Color Imaging System, Life Technologies).

## DNA constructs

The open reading frame (ORF) of murine *Tfcp2l1* was directly amplified from a mESCs cDNA library with mTfcp2l1_ORF_F and mTfcp2l1_ORF_R primers. The amplified *Tfcp2l1* ORF constructs were cloned into pENTR4 plasmid (Invitrogen) modified by including CMV early enhancer/chicken β actin (CAG) promoter, pENTR4-CAG (Heo *et al*, 2017), and then cloned into the pLEX307 lentiviral vector (Addgene plasmid 41392) using the Gateway Technology reaction to express the non-tagged protein. For overexpression of the Flag-tagged protein, they were cloned into pCMV_3Tag-1 vector (Agilent Technologies, Santa Clara CA). Murine *Cdk1* ORF (Cat # MMM1013-202767395, Open Biosystems, Pittsburgh, PA) was subcloned into pENTR4 CAG or pENTR4-CAG-2HA (including CAG promoter and two tandem HA epitope tags before ORF) plasmids. Human TFCP2L1 (Cat # MHS6278-202806269) or CDK1 (Cat # OHS1770-202320538) ORF constructs were purchased from Open Biosystems. Murine *Tfcp2l1* wild-type and *Tfcp2l1*[Q214L/K216E] variant cloned into PiggyBac vector (Ye *et al*, 2013) were kindly provided by Dr. Qi-Long Ying, University of Southern California. Murine *Tfcp2l1* or human *TFCP2L1* variants at threonine-177 were generated by site-directed mutagenesis (Intron, Seoul, Korea). Primers used for ORF cloning and the site-directed mutagenesis are listed in Appendix Table S8.

## Ectopic expression

mESCs that stably overexpress the Flag-tagged Tfcp2l1 proteins were established by transfection of the indicated plasmids using Lipofectamine 2000 (Invitrogen), followed by selection under 1 mg/ml G418 Geneticin (Invitrogen) for 2 weeks. For expressing non-tagged Tfcp2l1 WT or variant proteins, lentivirus containing the corresponding ORFs cloned into the pLEX307 lentiviral vector was produced using a four-plasmid transfection system (Invitrogen). The recombinant pseudo-lentiviral particles were concentrated using Lenti-X Concentrator kit (Clontech, Mountain View, CA), infected into murine ESCs using 6 μg/ml polybrene (Invitrogen), and assays were performed at 4 days after infection.

## RNA interference (RNAi)

For the RNAi-mediated gene knock-down (KD) assay, shRNAs designed to the indicated target were cloned into the pLenti6/Block-iT lentiviral vector (Invitrogen). Lentiviral delivery of these shRNAs was performed using similar procedures. The target sequences in each shRNA are listed in Appendix Table S9.

## Promoter activity assay

The murine *Nanog* promoter luciferase cloned into the pGL3-Basic plasmid (Ye *et al*, 2013) were kindly provided by Dr. Qi-Long Ying, University of Southern California. Luciferase reporter plasmid containing 6 tandem copies of the composite murine Oct4/Sox2 binding site of the *Fgf4* enhancer (6 × O/S luc, kindly provided from Lisa Dailey, Addgene plasmid # 69445) was used to detect the transcription activity of Oct-4/SOX-2 complexes. Each construct was co-transfected with a vector expressing β-galactosidase. After 24 h of transfection into ESCs using Lipofectamine 2000 (Invitrogen), promoter activity was measured using the luciferase assay kit (Promega) and normalized to an equivalent amount of β-galactosidase activity.

## Reprogramming assays

MEFs from B6;CBA-Tg(Pou5f1-EGFP)2Mnn/J mice (also known as OG2; purchased from The Jackson Laboratory, Bar Harbor, ME, USA) were transduced (5,000 cells) at passage 2 with FUW-SOKM (kindly provided by Dr. Jongpil Kim, Dongguk University, Seoul, Korea) lentiviruses in 24-well culture dishes. The lentiviruses containing *Tfcp2l1* wild-type (WT) or variant expressing open reading frame (ORF) as well as shRNA specific to *Tfcp2l1* were used to infect MEFs 2 days before SOKM induction with 6 μg/ml Polybrene (Invitrogen). For conversion into naïve pluripotency, OG2-EpiSCs were dissociated using 0.5 mM EDTA (Gibco) and plated on feeder layer or feeder-free in EpiSC medium, consisting of N2B27 media with 1% KnockOut Serum Replacement (Life Technologies), 12 ng/ml bFGF (Invitrogen), and 20 ng/ml Activin-A (Invitrogen). The next day, OG2-EpiSCs were cultured in 15% heat-inactivated fetal bovine serum (HyClone) and 1,000 U/ml ESGRO/LIF (Millipore) with the infection of the indicated lentiviruses.

## Statistics

*In vitro* molecular and cell biology and *in vivo* xenograft data from at least three independent replicates were analyzed by one-way or two-way analysis of variance (ANOVA) with Bonferroni *post hoc* tests. GraphPad Prism 7.0 (GraphPad Software, La Jolla, CA) was used for all analyses. Samples and animals were randomly allocated to the experimental groups and the order of treatment and evaluation. In animal studies, investigators involved in surgical procedures were blinded to the types of injected cells. All tumor size measurements and histological assessments were carried out by investigators who were blinded to the treatment groups.

For patients with BC, statistical analyses were performed using SPSS version 12.0. Pearson's chi-square and Fisher's exact tests were conducted to determine correlations between levels of antibody staining and clinicopathological parameters. Univariate survival analyses were performed to determine the clinical significance of protein expression and clinicopathological parameters. In multivariate analysis, a Cox proportional hazards model with the stepwise forward method was used to evaluate the hazard ratio for each factor, with a 95% confidence interval. All tests were two-sided, and *P*-values < 0.05 were considered statistically significant.

# Data availability

The datasets produced in this study are available in the following databases:

- ChIP-Seq dataset for Tfcp2l1 in mESCs: Gene Expression Omnibus GSE11431 (https://www.ncbi.nlm.nih.gov/geo/query/acc.cgi?acc = GSE11431).
- Mass spectrometry datasets: PRIDE PXD015635 (http://www.ebi.ac.uk/pride/archive/projects/PXD015635).
- MetaCore software was used for functional analysis of Tfcp2l1

transcription targets and interactome for core analyses of gene networks, biofunctions, and canonical pathways with default settings. Datasets used in MetaCore analysis are available in Datasets EV1 and EV2.

**Expanded View** for this article is available online.

## Acknowledgements

We thank Dr. Jae-Ouk Ahn at the Department of Medical IT Engineering, Soonchunhyang University, for statistical consultations. We thank Dr. Hans R. Scholer (Max Planck Institute for Molecular Biomedicine, Münster, Germany) for gcOct-4-GFP mESCs, Dr. Jongpil Kim (Dongguk University, Seoul, Korea) for the FUW-SOKM lentiviral construct, and Dr. Qi-Long Ying (University of Southern California, Los Angeles, CA, USA) for the constructs of murine wild-type *Tfcp2l1* gene cloned into PiggyBac vector, and *Nanog* promoter-driven luciferase reporter. This research was supported by a Basic Science Research Program through the National Research Foundation of Korea (NRF-2018R1A2B2001392), by a NRF MRC grant funded by the Korean government (MSIP) (2018R1A5A2020732), by the Korean Health Technology R&D Project, Ministry of Health & Welfare, Republic of Korea (HI18C2391), and by the Ministry of Education (2018R1D1A1B07047450 and 2017R1D1A1B03031379).

## Author contributions

Conceptualization, D-MS and YMC; Methodology, D-MS, YMC, JH, B-JN, and SL; Investigation, JH, B-JN, SL, H-YL, YHK, JL, HJ, HYY, C-MR, PCWL, HJ, YO, KK, JSK, and D-MS; Writing—original draft, D-MS, YMC, JH, B-JN, and SL; Writing—review and editing, D-MS, YMC, JS, and BH; Funding acquisition, D-MS; Resources, S-YK, JS, and BH; Data curation, D-MS and B-JN; Supervision, D-MS and YMC.

## Conflict of interest

The authors declare that they have no conflict of interest.

## For more information

(i) http://kmplot.com
(ii) https://cancergenome.nih.gov
(iii) ftp://asansdm.iptime.org/, FTP server for raw data, login ID/password (asan-guest/guest)

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
