## [Review Process File · EMBO Molecular Medicine]

Phosphorylation of TFPC2L1 by CDK1 is required for stem-cell pluripotency and bladder carcinogenesis

Jinbeom Heo, Byeong-Joo Noh, Seungun Lee, Hye-Yeon Lee, YongHwan Kim, Jisun Lim, Hyein Ju, Hwan Yeul Yu, Chae-Min Ryu, Peter C. W. Lee, Hwangkyo Jeong, Yumi Oh, Kyunggon Kim, Sang-Yeob Kim, Jaekyoung Son, Bumsik Hong, Jong Soo Kim, Yong Mee Cho, and Dong-Myung Shin

Review timeline:

Submission date:	13 May 2019
Editorial Decision:	14 June 2019
Revision received:	23 August 2019
Editorial Decision:	23 September 2019
Revision received:	10 October 2019
Accepted:	14 October 2019

Editor: Céline Carret

Transaction Report:

1st Editorial Decision

14 June 2019

Thank you for the submission of your manuscript to EMBO Molecular Medicine. We have now heard back from the three referees whom we asked to evaluate your manuscript.

You will see from the comments pasted below that all referees find the study interesting and clinically relevant for bladder cancer. Still, the referees have several suggestions that if followed, would strongly strengthen the study. While Ref. #1 has only minor issues, Ref. #2 and Ref. #3 have more pertinent questions, request explanations and more in vivo data to better match the in vitro experiments, in order to increase the conclusiveness of the paper.

At this stage, and given the reports, we would like to invite the submission of a revised version within three months for further consideration and would like to encourage you to address all the criticisms raised as suggested to improve conclusiveness and clarity. Please note that EMBO Molecular Medicine strongly supports a single round of revision and that, as acceptance or rejection of the manuscript will depend on another round of review, your responses should be as complete as possible.

Please also contact us as soon as possible if similar work is published elsewhere. If other work is published, we may not be able to extend the revision period beyond three months.

I look forward to receiving your revised manuscript.

***** Reviewer's comments *****

Referee #1 (Comments on Novelty/Model System for Author):

The model used are all appropriate

Referee #1 (Remarks for Author):

Some pluripotency-associated transcription factors like TFCEP2L1 are expressed in both early embryonic and adult tissues. Molecular programs involved in embryogenesis are frequently upregulated in oncogenic dedifferentiation and metastasis; however, their precise roles and regulatory mechanisms remain elusive. Here, the authors showed that phosphorylation of TFCEP2L1 by CDK1 orchestrated pluripotency and cell cycling in embryonic stem-cells (ESCs) and was aberrantly activated in aggressive bladder cancers (BCs). In murine ESCs, the protein interactome and transcription targets of Tfcpe2l1 were indicative of cell cycle regulation.

Tfcpe2l1 was phosphorylated at Thr177 by Cdk1, thereby regulating ESC cycle progression, pluripotency, and differentiation. The CDK1-TFCEP2L1 pathway was activated in human BC cells, stimulating their proliferation, self-renewal, and invasion capacities. Lack of TFCEP2L1 phosphorylation impaired tumorigenic potency of BC cells in a xenograft model. In patients with BC, a high level of co-expression of TFCEP2L1 and CDK1 was associated with unfavorable clinical characteristics including tumor grade, lymphovascular, muscularis propria invasion, and distant metastasis and served as an independent prognostic factor for cancer-specific survival. These findings demonstrate the molecular and clinical significance of CDK1-mediated phosphorylation of TFCEP2L1 in stem-cell pluripotency and BC progression, advancing our understanding of how transcription regulation related to normal stemness can modulate pathogenic stemness features.

The paper has strong clinical relevance for BC treatment. In general, the experiments are well presented and all controls appropriate. The conclusion is strongly supported by the data presented. There are some minor issues/clarifications required to further improve this work.

1) Page 15, 339 "Gene expression profiling analyses of MIBC cell lines have revealed an aggressive 340 basal-like subtype (5637 and HT1197) and a less aggressive luminal-like subtype (RT4 and HT1376)"

RT4 cells may be luminal-like but they are not representative of muscle-invasive bladder cancer. They are actually derived from a papillary (superficial) tumor.

2) The Discussion is very slim and the bladder cancer section very limited. Please expand considerably

Referee #2 (Comments on Novelty/Model System for Author):

The jump from ES cells to cancer is interesting, the choice of bladder cancer is justified but whether this represents really a bladder cancer-selective effect is dubious given the probably broad role of this protein in cell cycle regulation.

Referee #2 (Remarks for Author):

I don't find that the work provides an advance that is sufficiently relevant to the bladder cancer field (and it certainly does not to the ES cell field).

Heo et al report on a large series of studies in murine ES cells and human bladder cancer cells pointing to an important role of TFCEP2L1 in cell cycle and a pivotal role of Thr177 phosphorylation - mediated by CDK1 - in this process. The work focuses then on bladder cancer cells supporting that this protein, the kinase, and the phosphorylation event are important in tumorigenesis. Overall, the work is well performed although I have two major points to make: 1) the role of TFCEP2L1 in pluripotency was previously described and here no significant advances are made; 2) it appears clear that CDK1 can phosphorylate TFCEP2L1 (although no direct in vitro kinase is shown) and that this may be important for cell proliferation. No detailed analysis is performed of the cell cycle defects in cells carrying the T177A mutant in order to decipher the mechanisms involved and it is likely that

this cell cycle effect may contribute to most of the tumor phenotypes observed. I do not see strong evidence for a stem cell phenotype conferred by TFCEP2L1 from the studies presented. The significant association with survival in the multivariable Cox association study is poorly described since some of the strongest predictors from the univariable analysis (grade, lymphovascular invasion) do not appear to have been assessed in the multivariable analysis (which is not appropriate). In any case, the HR reported - in the range of 2 - is statistically significant but of borderline clinical relevance. Possibly Ki67 or other more common cell cycle markers might yield a similar HR.

Overall, I think that the current work - while well performed - does not provide a key novel finding in the field.

Referee #3 (Remarks for Author):

In this manuscript the authors describe their findings on posttranslational modification of TFCEP2L1. They find using proteomic analysis that TFCEP2L1 interacts with several pluripotency genes but also with cell cycle regulated genes containing the CDK1 and Wnt pathways. Further analysis revealed that TFCEP2L1 physically interacts with CDK1 and becomes phosphorylated and this regulates pluripotency. Importantly, this axis was also detected in bladder cancer and was crucial for xenograft growth and associated with poor prognosis.

Although the growth data are compelling in Figure 1 they do not match the cell cycle analyses. The authors show lack of growth over a period of 3 days and even regression, but the cell cycle distribution suggests that these cells are still cycling albeit a bit lower. This should be addressed in a more dynamic fashion. Importantly, the authors should also check the values as their G1? S/G2/M numbers only go up to 70% total. Is the rest dead?

The use of the T177E mutant has been done quite selectively. Although this mutant is clearly active and induces gene expression to a higher extent than the wt form (fig 1), it is not as effective in reprogramming while the effect on proliferation is not shown. These inconsistencies in use (why not shown in proliferation or colony formation or naive reprogramming) but also effect should be explained.

Finally, the authors should explain and study the phosphorylation of the wt form better. Overexpression of wt TFCEP2L1 results in clear effects and this is associated with phosphorylation of this form by CDK1. However, the authors also reveal that the wt form can rescue CDK1 knockdown. How does this work? Is phosphorylation not needed under these conditions? If so why not?

Fig 5D needs to be checked as the p-TFCEP2L1 is constant, this cannot be correct.

Finally, the in vivo data are less pronounced as one is made to believe. The expression of the TFCEP2L1 T177A mutant has strong effects in vitro as compared to the sham but hardly any effect is observed in vivo. This needs explanation.

1st Revision - authors' response

23 August 2019

Reviewer's comments

Response to Reviewer #1. We thank the reviewer for the positive appraisal of our study.

Referee #1 (Comments on Novelty/Model System for Author):

The model used are all appropriate

Response: We thank the reviewer for this encouraging comment. In this study, we examined the functional significance of TFCEP2L1 Thr177 phosphorylation by CDK1 in bladder carcinogenesis by performing a retrospective study of 400 bladder cancer patients and validated this phosphorylation as a critical regulator of embryonic and bladder cancer

phenotypes through a series of *in vitro* murine embryonic stem cell (mESC) and human bladder cancer cell line experiments, in addition to *in vivo* orthotopic xenograft assays.

Referee #1 (Remarks for Author):

Some pluripotency-associated transcription factors like TFCP2L1 are expressed in both early embryonic and adult tissues. Molecular programs involved in embryogenesis are frequently upregulated in oncogenic dedifferentiation and metastasis; however, their precise roles and regulatory mechanisms remain elusive. Here, the authors showed that phosphorylation of TFCP2L1 by CDK1 orchestrated pluripotency and cell cycling in embryonic stem-cells (ESCs) and was aberrantly activated in aggressive bladder cancers (BCs). In murine ESCs, the protein interactome and transcription targets of Tfc2l1 were indicative of cell cycle regulation. Tfc2l1 was phosphorylated at Thr177 by Cdk1, thereby regulating ESC cycle progression, pluripotency, and differentiation. The CDK1-TFCP2L1 pathway was activated in human BC cells, stimulating their proliferation, self-renewal, and invasion capacities. Lack of TFCP2L1 phosphorylation impaired tumorigenic potency of BC cells in a xenograft model. In patients with BC, a high level of co-expression of TFCP2L1 and CDK1 was associated with unfavorable clinical characteristics including tumor grade, lymphovascular, muscularis propria invasion, and distant metastasis and served as an independent prognostic factor for cancer-specific survival. These findings demonstrate the molecular and clinical significance of CDK1-mediated phosphorylation of TFCP2L1 in stem-cell pluripotency and BC progression, advancing our understanding of how transcription regulation related to normal stemness can modulate pathogenic stemness features.

The paper has strong clinical relevance for BC treatment. In general, the experiments are well presented and all controls appropriate. The conclusion is strongly supported by the data presented. There are some minor issues/clarifications required to further improve this work.

1) Page 15, 339 "Gene expression profiling analyses of MIBC cell lines have revealed an aggressive 340 basallike subtype (5637 and HT1197) and a less aggressive luminal-like subtype (RT4 and HT1376)" RT4 cells may be luminal-like but they are not representative of muscle-invasive bladder cancer. They are actually derived from a papillary (superficial) tumor.

Response: We apologize for this mistake. According to your suggestion, we now indicate that the RT4 cell line can be used as a model of non-muscular invasive bladder cancer (NMIBC) in the revised manuscript (Lines 340–342; Line 350; Line 382; Line924); (Nickerson, Witte et al., 2017)

2) The Discussion is very slim and the bladder cancer section very limited. Please expand considerably

Response: We thank the reviewer for this suggestion. In accordance with the reviewer's suggestion, we have added a section to the Discussion indicating that this type of cancer is a severe unmet medical problem with a high incidence and huge economic burden, and that molecular targeting therapies, such as those used for the treatment of other cancers, are in their infancy. Limited improvement in the clinical management of bladder cancer in recent decades has been attributed to the high level of clinical and pathological heterogeneity of bladder cancer. Therefore, it is important to investigate the molecular mechanisms of this type of cancer not only to advance our understanding of the pathogenesis of bladder cancer, but also to develop novel therapeutic strategies for the management of this aggressive disease, which has a high rate of recurrence and distant metastasis. We now mention this in the Introduction and Discussion of the revised manuscript (Lines 93–97; Lines 431–438). In addition, in response to the comments of reviewer #2, we have added a discussion about the significance of late cell-cycle activity, such as that involving the CDK1-TFCP2L1 pathway, on the pathogenesis and molecular characteristics of bladder cancer in the revised manuscript (Lines 454 – 474). We thank the reviewer for this helpful comment.

Response to Reviewer #2. We thank the reviewer for carefully evaluating our paper. The reviewer raised some concerns, which we address on a point-by-point basis below.

Referee #2 (Comments on Novelty/Model System for Author):

The jump from ES cells to cancer is interesting, the choice of bladder cancer is justified but whether this represents really a bladder cancer-selective effect is dubious given the probably broad role of this protein in cell cycle regulation.

Response: We thank the reviewer for this comment. In early embryonic developmental stage, TFCEP2L1 has a central role in the maintenance of a naïve state of pluripotency by interacting with many transcriptional regulators and chromatin-modifying complexes with roles in pluripotency. Unlike typical pluripotency TFs (OCT4 and NANOG), TFCEP2L1 is expressed in the epithelium of developing and adult organs, especially in the ducts of exocrine glands and the kidney. Accordingly, based on the database in the Genotype-Tissue Expression (GTEx) Portal (<https://gtexportal.org/home/>), high expression of TFCEP2L1 is observed in salivary gland and kidney as previously reported. Among adult tissues, the bladder shows high TFCEP2L1 expression (revised Fig EV3A). To explore whether TFCEP2L1 expression is associated with survival of patients with tumors derived from TFCEP2L1 expressing tissues, we employed open access databases on the Kaplan-Meier plotter (<http://kmplot.com/>) and The Cancer Genome Atlas (TCGA), particularly for datasets of kidney, stomach, esophagus, prostate, breast, and lung tissues in which the expression of TFCEP2L1 was higher than in the bladder (revised Fig EV3A–C). Among them, bladder and gastric cancer patients with higher *TFCEP2L1* expression were associated with shorter overall survival (revised Fig EV3B–D). Importantly, the expression level of *TFCEP2L1* in bladder cancer was more clinically useful than that of other stem cell markers (revised Fig EV3D). To make this clearer, we have presented these results as revised Fig EV3 and refer to them in the revised manuscript (Lines 274 – 281).

It should be stressed that aberrations in cell cycle regulation are one of the most extensively studied molecular aspects of bladder cancer (Kim, Akbani et al., 2015, Wang, Chen et al., 2019) and the high expression of late cell-cycle activity including those of CDK1/cyclin B complex or genes related to chromosome segregation and cell division is associated with subtypes of bladder cancer with poor prognosis (Hedegaard, Lamy et al., 2016, Sjobahl, Lauss et al., 2012). Therefore, late cell cycle activity, such as that of the CDK1-TFCEP2L1 pathway, could play a specific role in the pathogenesis and the related molecular features of bladder cancer, which should be explored by further investigation. To make this clearer, we have addressed this issue in the Discussion of the revised manuscript (Lines 454–474).

Referee #2 (Remarks for Author):

I don't find that the work provides an advance that is sufficiently relevant to the bladder cancer field (and it certainly does not to the ES cell field). Heo et al report on a large series of studies in murine ES cells and human bladder cancer cells pointing to an important role of TFCEP2L1 in cell cycle and a pivotal role of Thr177 phosphorylation - mediated by CDK1 – in this process. The work focuses then on bladder cancer cells supporting that this protein, the kinase, and the phosphorylation event are important in tumorigenesis. Overall, the work is well performed although

I have two major points to make:

1) the role of TFCEP2L1 in pluripotency was previously described and here no significant advances are made;

Response: We thank the reviewer for this comment. As the reviewer pointed out, the role of TFCEP2L1 in pluripotency has been previously reported; however, its regulatory mechanisms were not determined. In particular, post-translational modifications (PTM) including ubiquitination, SUMOylation, phosphorylation, methylation, and acetylation are important for regulation of the levels and activities of pluripotency-associated transcription factors (TF), to achieve a balance between pluripotency and differentiation. Modification of these TFs alters their transcriptional activities through regulation of DNA binding or protein stability (through ubiquitination). For example, the phosphorylation of Oct-4 at Ser229 by protein kinase A sterically hinders both Oct-4 DNA binding and homodimer assembly (Saxe, Tomilin et al., 2009). Phosphorylation of SOX-2 at Thr118 by protein kinase B (Jeong, Cho et al., 2010) promotes its stability by blocking ubiquitination, thus enhancing the self-renewal capacity of mESCs. However, Klf4 phosphorylation by extracellular signal-regulated kinase 1 (Erk1) or Erk2 at Ser123 leads to the recruitment of β TrCP1 or β TrCP2 (F-box proteins with E3 ubiquitin ligase activity) to the Klf4 N-terminal domain, resulting in ubiquitination and degradation of Klf4 (Kim, Kim et al., 2012). Here, we found that murine and human

TFCP2L1 proteins are phosphorylated by CDK1 at Thr177, which is located within the CP2-like domain that is responsible for DNA binding (Kim, Jang et al., 2016). Preventing this phosphorylation altered Tfc2l1 DNA binding and transcriptional activity of the specific binding targets. Genome-wide analysis of a previously reported Tfc2l1 ChIP-seq database, along with the results of our biochemical analysis, demonstrated that genes involved in processes related to the cell-cycle or tissue development were highly enriched among Tfc2l1 transcription targets and were affected by Thr177 phosphorylation. Accordingly, abnormal activity of the CDK1–TFCP2L1 pathway disturbed the regulation of cell-cycle and differentiation processes, resulting in impairment of pluripotency in mESCs. More importantly, we found that this molecular program for pluripotency is involved in the bladder carcinogenesis and proved the clinical relevance and utility of TFCP2L1 phosphorylation by performing a retrospective study of bladder cancer specimens from 400 patients at our institute and a TCGA dataset of bladder cancer patients. Taken together, these findings provide novel evidence for the molecular and clinical significance of CDK1-mediated phosphorylation of TFCP2L1 in stem cell pluripotency and bladder cancer genesis, progression and prognosis, advancing our understanding of how transcription regulation related to normal stemness can modulate pathogenic stemness features.

2) it appears clear that CDK1 can phosphorylate TFCP2L1 (although no direct in vitro kinase is shown) and that this may be important for cell proliferation. No detailed analysis is performed of the cell cycle defects in cells carrying the T177A mutant in order to decipher the mechanisms involved and it is likely that this cell cycle effect may contribute to most of the tumor phenotypes observed. I do not see strong evidence for a stem cell phenotype conferred by TFCP2L1 from the studies presented.

Response: We thank the reviewer for this comment. To address these concerns, we performed an *in vitro* kinase assay using human CDK1/Cyclin B recombinant proteins and a 19-mer peptide containing Thr177 that is conserved in TFCP2L1 proteins from all species examined (the revised Appendix Fig S1B). Mass spectrometry analysis of *in vitro* kinase products revealed that the TFCP2L1 peptide containing Thr177, but not the Thr177A mutant peptide was phosphorylated by human CDK1/Cyclin B recombinant proteins, indicating that CDK1 directly phosphorylates TFCP2L1 protein at Thr177. Furthermore, we found the human TFCP2L1 peptides phosphorylated at Thr177 by mass spectrometry analysis of IP analysis of FLAG-tagged TFCP2L1 in T24 cells, indicating Thr177 of TFCP2L1 is phosphorylated in human bladder cancer cells. To make this clearer, we have presented these results in revised Appendix Fig S5 and S6 and described them in the revised manuscript (Lines 373 – 377).

Regarding the phenotypes of bladder cancer cells carrying the T177A TFCP2L1 mutant, forced expression of T177A TFCP2L1 or shRNA (shTFCP2L1) led to significant depletion of bladder cancer T24 cells in S phase (revised Figs EV5I) and induced the differentiation of genes including those of bone morphogenetic protein (BMP), inhibitor of DNA binding (ID), and GATA binding family proteins (revised Figs EV5J). By contrast, cells with ectopic expression of wild-type or T177E TFCP2L1 showed up-regulation of cell-cycle related genes, as well as stemness and down-regulation of differentiation genes. These results provide additional evidence for the crucial role of TFCP2L1 Thr177 phosphorylation in generating stemness features of human bladder cancer cells. To make this clearer, we have presented these results in revised Fig EV5I and J and refer to them in the revised manuscript (Lines 382–393).

The significant association with survival in the multivariable Cox association study is poorly described since some of the strongest predictors from the univariable analysis (grade, lymphovascular invasion) do not appear to have been assessed in the multivariable analysis (which is not appropriate).

Response: We acknowledge the reviewer's concern. During statistical analysis, we included all the variables that were significant in the univariate analysis into the multivariate analysis including grade and lymphovascular invasion. The multivariate analysis was evaluated using the Cox proportional hazard model with the stepwise regression procedure, which revealed that age, muscularis propria invasion, lymph node metastasis, and high co-expression of p-TFCP2L1 and CDK1 still remained independent prognostic factors for bladder cancer-specific survival but not for grade or lymphovascular invasion (Chang, Cho et al., 2017,

Hachulla, Clerson et al., 2015). The results of the statistical analysis has been carefully examined by an expert statistician (Jae-Ouk Ahn MD, PhD at the Department of Medical IT Engineering, Soonchunhyang University, Republic of Korea). His contribution is acknowledged under Acknowledgements as follows.

Acknowledgements

We thank Dr. Jae-Ouk Ahn at the Department of Medical IT Engineering, Soonchunhyang University, for statistical consultations.

In any case, the HR reported - in the range of 2 - is statistically significant but of borderline clinical relevance. Possibly Ki67 or other more common cell cycle markers might yield a similar HR. Overall, I think that the current work - while well performed - does not provide a key novel finding in the field.

Response: As the reviewer pointed out, the hazard ratio was not that remarkable. Even those of well-known prognostic factors of muscularis propria invasion and lymph node metastasis are 2.464 and 2.527, respectively. In our previous study, we tested various proteins involved in the cell proliferation and cell cycle including Ki-67, p53, pRb, p27, and CDK4 in those cases (Kim, Sung et al., 2015). However, their hazard ratios for disease-specific survival were less than 1.5 as shown in the below table.

Protein	Disease-specific survival		
	HR	95% CI	P value
Ki-67	1.32	1.19–1.46	<0.001
p53	1.16	1.09–1.22	<0.001
pRb	1.44	1.24–1.65	<0.001
p27 (nucleus)	0.96	0.74–1.23	0.747
p27 (cytoplasm)	0.93	0.85–1.01	0.111
CDK4	1.07	0.75–1.53	0.701

Therefore, we think our results should be validated in separate studies with prospective design and a larger number of bladder cancer cases. We have mentioned this in the Discussion as a limitation as follows (Lines 509 – 512 in the revised manuscript).

In addition, to overcome the limitations of present study regarding its retrospective design and the relatively small number of cases, the significance of CDK1 and TFCP2L1 as prognostic markers need to be independently validated in separate studies with prospective design and a larger number of BC cases.

Response to Reviewer #3.

We thank the reviewer for carefully evaluating our paper.

Referee #3 (Remarks for Author):

In this manuscript the authors describe their findings on posttranslational modification of TFCP2L1. They find using proteomic analysis that TFCP2L1 interacts with several pluripotency genes but also with cell cycle regulated genes containing the CDK1 and Wnt pathways. Further analysis revealed that TFCP2L1 physically interacts with CDK1 and becomes phosphorylated and this regulates pluripotency. Importantly, this axis was also detected in bladder cancer and was crucial for xenograft growth and associated with poor prognosis.

Although the growth data are compelling in Figure 1 they do not match the cell cycle analyses. The authors show lack of growth over a period of 3 days and even regression, but the cell cycle distribution suggests that these cells are still cycling albeit a bit lower. This should be addressed in a more dynamic fashion. Importantly, the authors should also check the values as their G1? S/G2/M numbers only go up to 70% total. Is the rest dead?

Response: We thank the reviewer for this helpful comment. In accordance with the reviewer's suggestion, we analyzed daily changes in cell cycle progression and proliferation of mESCs after overexpression of *Tfcp2l1*-wild-type (WT), *Tfcp2l1*-T177A, or *Tfcp2l1*-T177E. In mESCs expressing T177A *Tfcp2l1*, the number of S phase cells decreased gradually on day 1 after infection with lentiviruses containing *Tfcp2l1*-T177A, decreased markedly on day 2, and decreased to a level that was half that of the number of control cells in S phase on day 3. Accordingly, defects in mESC proliferation were apparent 2 days after overexpression of the T177A *Tfcp2l1* variant or shRNA (*shTfcp2l1*) (the revised Fig 1J). To make this clearer, we have added the cell cycle analysis results at day 1 and day 2 days to revised Appendix Fig S3C and have replaced the data in original Fig 1K (cell cycle status in 3 days) with the new results. We have also replaced the data in original Fig 1J with the new results showing the positive effect of T177E *Tfcp2l1* expression on the proliferation of mESCs. The new results are referred to in the revised manuscript (Lines 195–200). We thank the reviewer for this important comment.

The use of the T177E mutant has been done quite selectively. Although this mutant is clearly active and induces gene expression to a higher extend than the wt form (fig 1), it is not as effective in reprogramming while the effect on proliferation is not shown. These inconsistencies in use (why not shown in proliferation or colony formation or naive reprogramming) but also effect should be explained.

Response: We apologize for these mistakes. We have added results showing the effect of T177E *Tfcp2l1* expression on cell proliferation (revised Fig 1J), the formation of undifferentiated alkaline phosphatase (AP)-positive ESC colonies (left panel in the revised Appendix Fig S3J), and also its effect on naïve reprogramming (revised Fig 2E). Expression of the *Tfcp2l1* phospho-mimic T177E variant increased cell proliferation, cell cycle progression, as observed with wild-type *Tfcp2l1*, and induced a subset of *Tfcp2l1*-targeted genes, which are annotated by cell cycle GO terms (revised Figs 1J–L and Appendix Fig S3J). Naïve reprogramming was little affected by the expression of the T177E *Tfcp2l1* variant, unlike the stimulation of naïve reprogramming by *Tfcp2l1* WT, but it was significantly repressed by the T177A *Tfcp2l1* variant. To make this clearer, we have revised the original Figs 1J and 2E by adding the results for *Tfcp2l1* T177E variant and refer to these results in the revised manuscript (Lines 199–200; Lines 246–248).

In several proliferation (revised Fig 1J) and differentiation (revised Fig 2A–D) assays of mESCs, expression of T177E *Tfcp2l1* variant showed less biological activity than wild-type *Tfcp2l1*. Additionally, it had little effect on the stemness features of human bladder cancer cell-lines (revised Fig 6G–J). Expression of this phospho-mimic construct might interfere with other unidentified crucial post-translational modifications (PTM; e.g., acetylation, ubiquitination, and SUMOylation) of TFPC2L1 or it might have effects on other transcription regulators. Therefore, further investigation is required to elucidate the complexity of the TFPC2L1-centered transcription factor network, the components and regulators of which could critically determine the transcription activity of TFPC2L1. To make this clearer, we have addressed this issue in detail in the Discussion of the revised manuscript (Lines 466–474).

Finally, the authors should explain and study the phosphorylation of the wt form better. Overexpression of wt TFPC2L1 results in clear effects and this is associated with phosphorylation of this form by CDK1. However, the authors also reveal that the wt form can rescue CDK1 knockdown. How does this work? Is phosphorylation not needed under these conditions? If so why not?

Response: We thank the reviewer for this critical comment. CDK1 has pleiotropic effects via its ability to modify a variety of substrates (Petrone, Adamo et al., 2016). CDK1 regulates ES cell survival through the p53-NOXA-MCL1 pathway (Holt, Tuch et al., 2009, Huskey, Guo et al., 2015). Accordingly, when we examined cells carrying five independent *CDK1* shRNA constructs, we found that severe knock-down of *CDK1* (*shRNA#1*) led to rapid cell death of both mESCs (revised Appendix Fig S3H and I) and bladder cancer cells (revised Fig EV5C and D). Thus, a *shCDK1* construct (*shRNA#2*) with moderate silencing activity was used to investigate whether expression of wild type TFPC2L1 could rescue the phenotypes of mESCs and bladder cancer cells with the reduced *CDK1* expression. We found that TFPC2L1 phosphorylation was partially restored in these cells with moderate silencing of *CDK1* (revised Fig EV5E). To make this clearer, we have presented these results in revised Appendix Fig S3H

and I for mESCs and revised Fig EV5C–E for human T24 bladder cancer cells and describe the results in the revised manuscript (Lines 205–207; Lines 363–365). We thank the reviewer for this important comment.

Fig 5D needs to be checked as the p-TFCP2L1 is constant, this cannot be correct.

Response: We apologize for this mistake. We have replaced the representative p-TFCP2L1 western blot image in revised Fig 5D. In line with the increases in cell proliferation (revised Fig 5C) and stemness features (revised Fig 5F and H), T24 cells with ectopic expression of TFCP2L1 alone or with co-expression of TFCP2L1 and CDK1 showed an increased level of p-TFCP2L1 protein.

Finally, the in vivo data are less pronounced as one is made to believe. The expression of the TFCP2L1 T177A mutant has strong effects in vitro as compared to the sham but hardly any effect is observed in vivo. This needs explanation.

Response: Functional significance of TFCP2L1 Thr177 phosphorylation in bladder carcinogenesis was validated using *in vivo* xenograft models in which tumorigenicity of T24 cells overexpressing TFCP2L1 (wild-type or T177A/T177E variants) or *TFCP2L1* shRNA were examined after transplanting them orthotopically through the outer layer of the bladder of immunodeficient mice. Based on the tumor growth and histological findings, T24 cells carrying T177A *TFCP2L1* or sh*TFCP2L1* constructs showed severe defects in tumor formation (revised Fig 7B – D), unlike cells carrying the empty, wild-type TFCP2L1, or T177E TFCP2L1 constructs. Accordingly, majority of T24 cells expressing T177A *TFCP2L1* variant or sh*TFCP2L1* showed degeneration around the injection sites (revised Fig 7D). In line with these results, xenograft tumors of these cells showed minimal expression of p-TFCP2L1 and CDK1 proteins (revised Fig 7E), like the sham operated mice group, as well lower expression of bladder cancer stem cell markers, including CD44 (revised Fig 7F), KRT14, and SALL4 (Appendix Fig S8). To make this clearer, we have added arrowed lines to clearly mark each group in revised Figure 7B, which contains representative images of bladders bearing tumors 4 weeks after transplantation of the indicated T24 cells. In addition, we have added the immunofluorescent staining results for KRT14 and SALL4 to revised Appendix Fig 8 and described these results in the revised manuscript (Lines 415–416).

Please note that we have added two new co-authors, Hwangkyo Jeong and Yumi Oh, who performed mass spectrometry analysis of the phosphorylation sites of human TFCP2L1 proteins and *in vitro* kinase analysis during the revision period.

REFERENCES

- Chang CC, Cho SF, Tu HP, Lin CY, Chuang YW, Chang SM, Hsu WL, Huang YF (2017) Tumor and bone marrow uptakes on [18F]fluorodeoxyglucose positron emission tomography/computed tomography predict prognosis in patients with diffuse large B-cell lymphoma receiving rituximab-containing chemotherapy. *Medicine (Baltimore)* 96: e8655
- Hachulla E, Clerson P, Airo P, Cuomo G, Allanore Y, Caramaschi P, Rosato E, Carreira PE, Riccieri V, Sarraco M, Denton CP, Riemekasten G, Pozzi MR, Zeni S, Mihai CM, Ullman S, Distler O, Rednic S, Smith V, Walker UA et al. (2015) Value of systolic pulmonary arterial pressure as a prognostic factor of death in the systemic sclerosis EUSTAR population. *Rheumatology (Oxford)* 54: 1262-9
- Hedegaard J, Lamy P, Nordentoft I, Algaba F, Hoyer S, Ulhoi BP, Vang S, Reinert T, Hermann GG, Mogensen K, Thomsen MBH, Nielsen MM, Marquez M, Segersten U, Aine M, Hoglund M, Birkenkamp-Demtroder K, Fristrup N, Borre M, Hartmann A et al. (2016) Comprehensive Transcriptional Analysis of Early-Stage Urothelial Carcinoma. *Cancer Cell* 30: 27-42
- Holt LJ, Tuch BB, Villen J, Johnson AD, Gygi SP, Morgan DO (2009) Global analysis of Cdk1 substrate phosphorylation sites provides insights into evolution. *Science* 325: 1682-6
- Huskey NE, Guo T, Evason KJ, Momcilovic O, Pardo D, Creasman KJ, Judson RL, Billelloch R, Oakes SA, Hebrok M, Goga A (2015) CDK1 inhibition targets the p53-NOXA-MCL1 axis, selectively kills embryonic stem cells, and prevents teratoma formation. *Stem Cell Reports* 4: 374-89

- Jeong CH, Cho YY, Kim MO, Kim SH, Cho EJ, Lee SY, Jeon YJ, Lee KY, Yao K, Keum YS, Bode AM, Dong Z (2010) Phosphorylation of Sox2 Cooperates in Reprogramming to Pluripotent Stem Cells. *STEM CELLS* 28: 2141-2150
- Kim CM, Jang T-h, Park HH (2016) Functional Analysis of CP2-Like Domain and SAM-Like Domain in TFCP2L1, Novel Pluripotency Factor of Embryonic Stem Cells. *Applied Biochemistry and Biotechnology* 179: 650-658
- Kim J, Akbani R, Creighton CJ, Lerner SP, Weinstein JN, Getz G, Kwiatkowski DJ (2015) Invasive Bladder Cancer: Genomic Insights and Therapeutic Promise. *Clin Cancer Res* 21: 4514-24
- Kim K, Sung CO, Park BH, Ku JY, Go H, Ro JY, Kim J, Cho YM (2015) Immunoprofile-based subgrouping of urothelial bladder carcinomas for survival prediction. *Hum Pathol* 46: 1464-70
- Kim MO, Kim S-H, Cho Y-Y, Nadas J, Jeong C-H, Yao K, Kim DJ, Yu D-H, Keum Y-S, Lee K-Y, Huang Z, Bode AM, Dong Z (2012) ERK1 and ERK2 regulate embryonic stem cell self-renewal through phosphorylation of Klf4. *Nat Struct Mol Biol* 19: 283-290
- Nickerson ML, Witte N, Im KM, Turan S, Owens C, Misner K, Tsang SX, Cai Z, Wu S, Dean M, Costello JC, Theodorescu D (2017) Molecular analysis of urothelial cancer cell lines for modeling tumor biology and drug response. *Oncogene* 36: 35-46
- Petrone A, Adamo ME, Cheng C, Kettenbach AN (2016) Identification of Candidate Cyclin-dependent kinase 1 (Cdk1) Substrates in Mitosis by Quantitative Phosphoproteomics. *Mol Cell Proteomics* 15: 2448-61
- Saxe JP, Tomilin A, Schöler HR, Plath K, Huang J (2009) Post-Translational Regulation of Oct4 Transcriptional Activity. *PLoS ONE* 4: e4467
- Sjodahl G, Lauss M, Lovgren K, Chebil G, Gudjonsson S, Veerla S, Patschan O, Aine M, Ferno M, Ringner M, Mansson W, Liedberg F, Lindgren D, Hoglund M (2012) A molecular taxonomy for urothelial carcinoma. *Clin Cancer Res* 18: 3377-86
- Wang L, Chen S, Luo Y, Yuan L, Peng T, Qian K, Liu X, Xiao Y, Wang X (2019) Identification of several cell cycle relevant genes highly correlated with the progression and prognosis of human bladder urothelial tumor. *J Cell Physiol* 234: 13439-13451

2nd Editorial Decision

23 September 2019

Thank you for the submission of your revised manuscript to EMBO Molecular Medicine. We have now received the enclosed reports from the referees that were asked to re-assess it. As you will see the reviewers are now globally supportive and I am pleased to inform you that we will be able to accept your manuscript pending minor editorial amendments.

I look forward to reading a new revised version of your manuscript as soon as possible and within 2 weeks.

***** Reviewer's comments *****

Referee #2 (Comments on Novelty/Model System for Author):

The authors have performed the study with adequate controls and the results are convincing.

Referee #2 (Remarks for Author):

The authors have responded to my queries satisfactorily. I don't think that this would be a high priority paper for EMBO Mol Med readership, but I understand that the other two reviewers do not have the same opinion. From a technical standpoint, I do not need to see the paper again.

Referee #3 (Remarks for Author):

The authors have successfully addressed the concerns and have added sufficient novel information and data.

Authors made the requested changes.

Corresponding Author Name: Dong-Myung Shin and Yong Mee Cho

Manuscript Number: EMM-2019-10880-V2